# VERTIBENCH: ADVANCING FEATURE DISTRIBUTION DIVERSITY IN VERTICAL FEDERATED LEARNING BENCHMARKS

**Zhaomin Wu, Junyi Hou, Bingsheng He**
National University of Singapore
`{zhaomin,junyi.h,hebs}@comp.nus.edu.sg`

## ABSTRACT

Vertical Federated Learning (VFL) is a crucial paradigm for training machine learning models on feature-partitioned, distributed data. However, due to privacy restrictions, few public real-world VFL datasets exist for algorithm evaluation, and these represent a limited array of feature distributions. Existing benchmarks often resort to synthetic datasets, derived from arbitrary feature splits from a global set, which only capture a subset of feature distributions, leading to inadequate algorithm performance assessment. This paper addresses these shortcomings by introducing two key factors affecting VFL performance - feature importance and feature correlation - and proposing associated evaluation metrics and dataset splitting methods. Additionally, we introduce a real VFL dataset to address the deficit in image-image VFL scenarios. Our comprehensive evaluation of cutting-edge VFL algorithms provides valuable insights for future research in the field.

## 1 INTRODUCTION

Federated learning (Konečný et al., 2016) is acknowledged for enabling model training on distributed data with enhanced privacy. In this study, we delve into the less explored vertical federated learning (VFL), where each party has a feature subset, aligning with a general definition of federated learning (Li et al., 2021a) that includes privacy-preserving collaborative learning like assisted learning (Diao et al., 2022) and split learning (Vepakomma et al., 2018). The VFL application, depicted in Figure 1a, involves an initial development phase using synthetic or real-world benchmarks, followed by deployment in actual federated environments upon validation.

Evaluating VFL algorithms is challenging due to the inherent confidentiality of VFL data (Liu et al., 2022). The scope of party imbalance and correlation in existing real VFL datasets, termed the *real scope*, is limited. Datasets in the OARF benchmark (Hu et al., 2022), FedAds (Wei et al., 2023), NUS-WIDE (Chua et al., 2009), and Vehicle (Duarte and Hu, 2004), predominantly represent scenarios where parties are balanced and exhibit weak correlations, as depicted in Figure 1b.

To address the constraints inherent in the real scope, many VFL benchmarks (Hu et al., 2022; He et al., 2020; Caldas et al., 2018) utilize synthetic datasets. This evaluation scope, termed *uniform scope*, represent the imbalance-correlation scope under an equal distribution of features among parties, either randomly or manually. The uniform scope, though commonly adopted in VFL experiments (Diao et al., 2022; Castiglia et al., 2022), confines the evaluation to scenarios featuring balanced, strongly correlated parties according to Figure 1b. Another critical limitation is the misalignment between the uniform scope and real scope, underscoring the imperative for a diverse and realistic VFL benchmark.

Constructing a systematic synthetic VFL benchmark necessitates pinpointing the key factors affecting VFL algorithm performance. Existing synthetic benchmarks for non-i.i.d. *horizontal federated learning* (HFL), such as NIID-Bench (Li et al., 2022a), fall short for VFL due to inherent assumptions about feature space and instance significance. Specifically, while HFL benchmarks typically assume independent and uniformly significant instances, this does not hold in VFL where features exhibit intrinsic correlations and differing importances. Furthermore, HFL benchmarks posit that all parties share the same feature space, a premise misaligned with VFL's distributed feature paradigm. This delineates the unique analytical challenges inherent to synthetic VFL benchmarks.

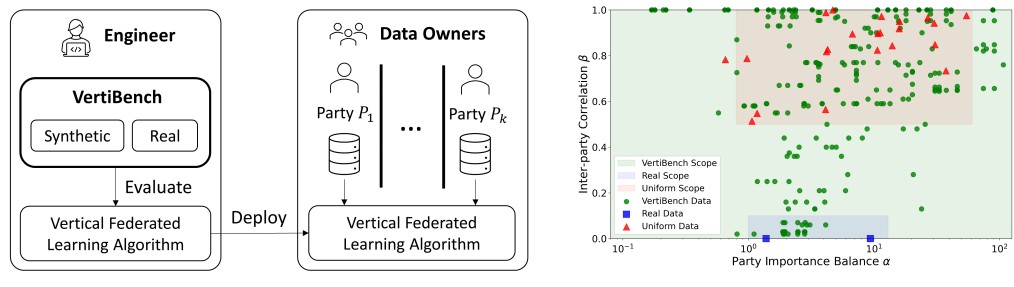

(a) Pipeline of VFL        (b) Estimated scope of VFL datasets

Figure 1: Overview of existing VFL piplines and datasets and the estimated scope of VFL datasets

Given these limitations, our statistical analysis of supervised VFL tasks identifies party importance and correlation as two crucial factors influencing target probability distributions in synthetic VFL datasets derived from the same global dataset. Accordingly, we propose *VertiBench*, a comprehensive VFL benchmark featuring novel feature-splitting methods for synthetic dataset generation. VertiBench offers three primary benefits: (1) it generally encompasses the uniform scope; (2) it effectively emulates the real scope, as evidenced by comparable performance on VertiBench-synthetic datasets; and (3) it introduces the capability to evaluate other scenarios that have not been explored in the previous studies, e.g. imbalanced feature split, broadening the scope of VFL evaluation.

Our primary contributions include: (1) Synthetic dataset generation methods with varied party importance and correlation, capturing a broad scope of VFL scenarios. (2) Novel real-world image-to-image VFL dataset `Satellite`. (3) Techniques to evaluate the party importance and correlation of real-world VFL datasets, enabling feature split comparison with synthetic VFL datasets. (4) Comprehensive benchmarks of mainstream cutting-edge VFL algorithms, providing key insights. For example, we demonstrate the scalability of VFL algorithms, challenging prior assumptions about VFL scaling difficulties (Hu et al., 2022), and emphasize the challenges of communication efficiency in VFL datasets across varying imbalance levels. The VertiBench source code is available on GitHub (Wu et al., 2023a), with data splitting tools installable from PyPI (Wu et al., 2023b). The pre-split dataset is accessible in (Anonymized, 2023).

## 2 EVALUATE VFL DATASETS

In this section, our objective is to investigate the primary factors influencing VFL performance when generating synthetic VFL datasets from a fixed global dataset. Additionally, we explore methods to efficiently estimate these factors, guiding the subsequent feature split.

### 2.1 FACTORS THAT AFFECT VFL PERFORMANCE

Suppose there are $K$ parties. Denote the data on party $P_k$ as a random vector $\mathbf{X}_k$ ($1 \leq k \leq K$). Denote the label as a random variable $y$. A supervised learning algorithm maximizes the likelihood function where hypothesis $h$ represents models and parameters, i.e., $L(y|\mathbf{X}_K, ..., \mathbf{X}_1; h)$. These supervised learning algorithms estimate the probability mass function in Eq. 1. The proof of Proposition 1 is provided in Appendix A.

**Proposition 1.** *The probability mass function can be written as*

$$\log \mathbb{P}(y|\mathbf{X}_K, ..., \mathbf{X}_1) = \sum_{k=1}^{K} \log \frac{\mathbb{P}(y|\mathbf{X}_k, ..., \mathbf{X}_1)}{\mathbb{P}(y|\mathbf{X}_{k-1}, ..., \mathbf{X}_1)} + \log \mathbb{P}(y) \tag{1}$$

In VFL, $\mathbb{P}(y)$ is the same for all the parties. The skewness among $K$ parties is determined by $K$ ratios of distributions. Interestingly, this ratio quantifies the divergence between two marginal probability distributions of $y$ - one inclusive of $\mathbf{X}_k$ and the other exclusive of $\mathbf{X}_k$. Essentially, the ratio estimates the impact on the global distribution when the features of a single party are excluded. This can be interpreted as the **importance** of a given party. Proposition 1 applies regardless of the order of $\mathbf{X}_1, ..., \mathbf{X}_k$. Shapley value, emphasizing feature independence, aids in precisely evaluating party importance in vertical federated learning, as demonstrated in (Wang et al., 2019; Han et al., 2021).

In another aspect, the ratio $\frac{\mathbb{P}(y|\mathbf{X}_k,\ldots,\mathbf{X}_1)}{\mathbb{P}(y|\mathbf{X}_{k-1},\ldots,\mathbf{X}_1)}$ is determined by the **correlation** between $\mathbf{X}_k$ and $\mathbf{X}_1,\ldots,\mathbf{X}_{k-1}$. In cases where the independence assumption underlying the Shapley value is invalidated, assessing each party's impact on the global distribution becomes more accurate when based on feature correlation.

We identify feature importance and correlation as pivotal factors influencing VFL algorithm performance. For datasets with nearly independent features, the low inter-party correlation makes correlation-based splits less meaningful, suggesting the superiority of importance-based feature splits. Conversely, in datasets with highly correlated features, assessing individual feature importance becomes impractical, making correlation-based splits more suitable due to varying inter-party correlations.

Importance and correlation are treated as orthogonal evaluation factors applicable in distinct scenarios. While there may be an intrinsic link between them, our experiments indicate that focusing on one factor at a time yields explainable results reflective of real-world performance. As discussed in Appendix H, the interplay between importance and correlation can be complex. A joint optimization for both factors might be computationally intensive and less explainable, while providing limited additional insights. The subsequent sections will introduce our approach to evaluate these two factors and generating synthetic datasets based on each factor accordingly.

## 2.2 Evaluate Party Importance

To assess the importance for each party, we sum the importance of its features. While numerous methods to evaluate feature importance can be adopted in VertiBench, this study primarily focuses on two approaches: 1) Shapley Value: Feature importance is determined using Shapley values, efficiently estimated by evaluating the performance of a trained XGBoost (Chen and Guestrin, 2016) on random subsets. 2) Shapley-CMI (Han et al., 2021): This approach, which does not rely on specific models, estimates the importance of each feature based on the Shapley-CMI applied to the global dataset. Both methods yield consistent and reasonable estimates of party importance.

## 2.3 Evaluate Party Correlation

The task of efficiently evaluating correlation among two groups of features is challenging despite well-studied individual feature correlation (Myers and Sirois, 2004; De Winter et al., 2016). The Shapley-Taylor index, proposed for evaluating correlation between feature sets (Sundararajan et al., 2020), is computationally intensive (NP-hard), and unsuitable for high-dimensional datasets. The determinant of the correlation matrix (Wang and Zheng, 2014) efficiently estimates inter-party correlation but is over-sensitive to linearly correlated features, impeding its use in feature partitioning. A more refined metric - the multi-way correlation coefficient (mcor) (Taylor, 2020), addresses this, but like the determinant, it struggles with unequal feature numbers across parties, a typical VFL scenario, due to the assumption of a square correlation matrix.

Given the limitations of existing metrics (Taylor, 2020; Wang and Zheng, 2014), we propose a novel metric to examine the correlation when the parties involved possess unequal numbers of features. Our approach hinges on the use of the standard variance of the singular values of the correlation matrix. This serves as an efficient measure of the overall correlation between two parties. Since the feature-wise correlation is an orthogonal research area, we selected Spearman rank correlation (Zar, 2005) due to its capability to handle non-linear correlation.

To elaborate further, we denote the column-wise correlation matrix between two matrices, $\mathbf{X}_i$ and $\mathbf{X}_j$, as $\text{cor}(\mathbf{X}_i, \mathbf{X}_j)$. As a result, we formally define the correlation between two entities, $\mathbf{X}_i \in \mathbb{R}^{n \times m_i}$ and $\mathbf{X}_j \in \mathbb{R}^{n \times m_j}$, in terms of their respective parties as Eq. 2.

$$\text{Pcor}(\mathbf{X}_i, \mathbf{X}_j) := \tfrac{1}{\sqrt{d}} \sqrt{\tfrac{1}{d-1} \textstyle\sum_{t=1}^{d} \left(\sigma_t(\text{cor}(\mathbf{X}_i, \mathbf{X}_j)) - \overline{\sigma}\right)^2}, \quad d = \min(m_i, m_j) \quad (2)$$

In this equation, $\sigma_i(\cdot)$ means the $i$-th singular value of a matrix, while $\overline{\sigma}$ stands for their mean value. Proposition 2 states that Pcor is equivalent to mcor for inner-party correlation (see Appendix A for proof). Experiments detailed in Appendix D.1 reveal that Pcor exhibits trends analogous to mcor (Taylor, 2020) when assessing inter-party correlation between equal number of features.

**Proposition 2.** *For any real matrix* $\mathbf{X}$, *Pcor*$(\mathbf{X}, \mathbf{X}) = mcor(\mathbf{X}, \mathbf{X})$

The singular values of a correlation matrix, Pcor, represent the magnitudes of its ellipsoid's semi-axes, indicating the degree of dependence among features. The standard deviation of these singular values reflects the distribution of dependence across different axes. A notably large singular value in a specific axis (Figure 2c) suggests a high concentration of dependence. For instance, if there's only one nonzero singular value, it implies that all features are perfectly correlated with a single feature. Conversely, if the singular values are uniformly distributed such as Figure 2a (indicated by a small standard deviation), it denotes less concentrated feature correlations. Therefore, the standard deviation of singular values serves as a measure of the dataset's proximity to perfect correlation.

Proposition 3 states that Pcor, like mcor, spans a range from 0 to 1, even when assessing inter-party correlation. A Pcor value of 1 signifies perfect correlation between $\mathbf{X}_1$ and $\mathbf{X}_2$, while a value of 0 indicates their independence.

**Proposition 3.** *For any two real matrices* $\mathbf{X}_1$ *and* $\mathbf{X}_2$, *Pcor*$(\mathbf{X}_1, \mathbf{X}_2) \in [0, 1]$

It is important to note that the absolute value of Pcor alone does not fully capture inter-party correlation. For instance, when $\mathbf{X}_i$ and $\mathbf{X}_j$ are two parties both containing the same set of independent features, Pcor$(\mathbf{X}_i, \mathbf{X}_j)$ yields a value of 0, the same as the Pcor between two independent parties. Despite the same Pcor value, these scenarios intuitively differ in their levels of inter-party correlation. This discrepancy arises from overlooking the inner-party correlation of $\mathbf{X}_i$ and $\mathbf{X}_j$. Typically, parties with highly correlated features tend to exhibit higher Pcor values with other parties.

To accurately measure the correlation between $\mathbf{X}_i$ and $\mathbf{X}_j$, we evaluate how the shift towards perfect correlation varies when $\mathbf{X}_i$ is replaced by $\mathbf{X}_j$. This is captured by the relative change in Pcor, denoted as Pcor$(\mathbf{X}_i, \mathbf{X}_j) -$ Pcor$(\mathbf{X}_i, \mathbf{X}_i)$. In the perspective of variance analysis (Kruskal and Wallis, 1952), this difference quantifies the degree to which the standard deviation Pcor$(\mathbf{X}_i, \mathbf{X}_j)$ is explained by inter-party factors, controlling the contribution of inner-party correlations. The overall inter-party correlation, denoted as Icor, is described as the mean party-wise correlation across all distinct party pairs. Formally,

$$\text{Icor}(\mathbf{X}_1, \dots, \mathbf{X}_K) := \frac{1}{K(K-1)} \sum_{i=1}^{K} \sum_{j=1, j \neq i}^{K} \left(\text{Pcor}(\mathbf{X}_i, \mathbf{X}_j) - \text{Pcor}(\mathbf{X}_i, \mathbf{X}_i)\right). \quad (3)$$

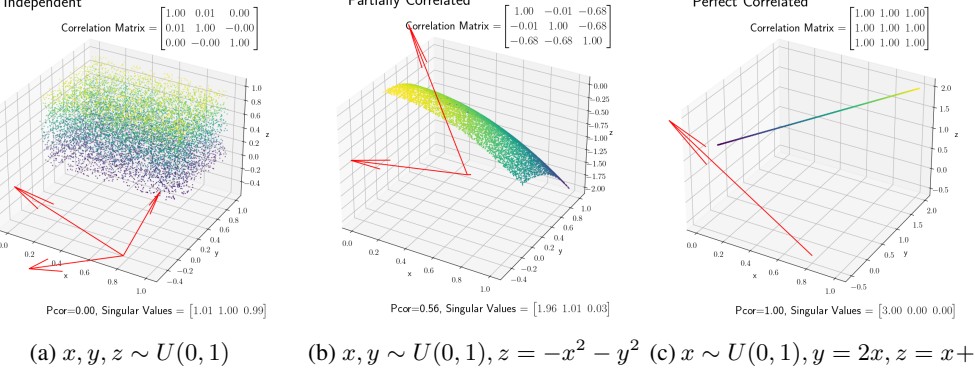

(a) $x, y, z \sim U(0, 1)$     (b) $x, y \sim U(0, 1), z = -x^2 - y^2$   (c) $x \sim U(0, 1), y = 2x, z = x + 1$

Figure 2: Examples of Pcor values on different levels of correlation. $U$ means uniform distribution. Arrow direction indicates right singular vector orientation, arrow scale represents singular values.

Icor exhibits notable properties both theoretically and empirically. Theoretically, as demonstrated in Theorem 1 (see Appendix A for proof), optimizing Icor yields ideal feature splits in optimal scenarios. Specifically, in datasets comprising two independent but internally perfectly correlated feature sets, Icor reaches its minimum when each party exclusively possesses one feature set and attains its maximum when each party equally shares half of the features from both sets. Empirically, we evaluate the link between inter-party correlation and Icor in complex, real-world datasets (Appendix D). These empirical observations align with theoretical insights, confirming Icor's capability in analyzing intricate data correlations.

**Theorem 1.** *Consider a global dataset* $\mathbf{X}$ *comprising two independent datasets* $\mathbf{D}_1, \mathbf{D}_2 \in \mathbb{R}^{n \times m}$, *each of the same dimension. Independence implies that for any feature* $a_i^{(1)}$ *from* $\mathbf{D}_1$ *and any feature* $a_j^{(2)}$ *from* $\mathbf{D}_2$, *where* $i, j \in [1, m]$, *the correlation* $Cor\left(a_i^{(1)}, a_j^{(2)}\right) = 0$. *Furthermore, assume within* $\mathbf{D}_1$ *and* $\mathbf{D}_2$, *all features are perfectly correlated, such that for all pairs of distinct features* $a_i^{(1)}, a_j^{(1)}$ *in* $\mathbf{D}_1$ *and* $a_i^{(2)}, a_j^{(2)}$ *in* $\mathbf{D}_2$, *with* $i, j \in [1, m]$ *and* $i \neq j$, *the correlations satisfy* $Cor\left(a_i^{(1)}, a_j^{(1)}\right) = 1$ *and* $Cor\left(a_i^{(2)}, a_j^{(2)}\right) = 1$ *respectively. When the features of* $\mathbf{X}$ *are divided equally into two subsets,* $\mathbf{X}_1$ *and* $\mathbf{X}_2$, *such that each subset contains* $m/2$ *features, the overall inter-party correlation* $Icor(\mathbf{X}_1, \mathbf{X}_2)$ *satisfies*

$$Icor(\mathbf{X}_1, \mathbf{X}_2) \in \left[-\frac{m}{\sqrt{m(m-1)}}, 0\right].$$

*The lower bound occurs if and only if* $\mathbf{X}_1$ *comprises all features of either* $\mathbf{D}_1$ *or* $\mathbf{D}_2$, *with* $\mathbf{X}_2$ *containing the remaining features. The upper bound occurs if and only if* $\mathbf{X}_1$ *holds* $m$ *features from both* $\mathbf{D}_1$ *and* $\mathbf{D}_2$, *with* $\mathbf{X}_2$ *holding the remaining* $m$ *features from* $\mathbf{D}_1$ *and* $\mathbf{D}_2$.

## 3 SPLIT SYNTHETIC VFL DATASETS

This section aims to develop algorithms to split features according to two key factors: importance and correlation. These algorithms should allow users to adjust the party importance and correlation of synthetic VFL datasets by simply modulating two parameters: $\alpha$ and $\beta$. The intended mapping should meet two criteria: (1) The scope of $\alpha$ and $\beta$ should encompass a broad spectrum of feature splits, inclusive of both real splits and random splits. (2) When two global datasets bear similarities, synthetic VFL datasets derived from them using identical $\alpha$ and $\beta$ parameters should yield similar VFL algorithm behaviors. We provide both theoretical and empirical validation for criteria (1) in this section, whereas criteria (2) is substantiated through experiments in Section 4.4.

### 3.1 SPLIT BY PARTY IMPORTANCE

In light of the computational expense incurred by the Shapley value method, an alternative and more efficient strategy is necessary to perform feature splits based on importance. With all parties exhibiting symmetry in the context of $\mathbf{X}$, varying the importance among parties essentially translates to varying the variance of the importance among them. Assuming each party $P_i$ possesses an importance factor $\alpha_i > 0$, we propose the implementation of the Dirichlet distribution parameterized by $\alpha = \{\alpha_i\}_{i=1}^K$ for feature splitting. This approach ensures two beneficial properties post-split: (1) a larger $\alpha_i$ guarantees a higher expected importance for $P_i$, and (2) a smaller $\|\{\alpha_i\}_{i=1}^K\|_2$ assures a greater variance in the importance among parties.

More specifically, we propose a feature splitting method based on feature importance. After initializing local datasets for each party, a series of probabilities $r_1, \ldots, r_K$ s.t. $\sum_{i=1}^K r_i = 1$ is sampled from a Dirichlet distribution $\text{Dir}(\alpha_1, \ldots, \alpha_K)$. Each feature is randomly allocated to a party $P_k$, selected based on the probabilities $r_k$. To accommodate algorithms that fail when faced with empty features, we can ensure each party is initially provided with a random feature before the algorithm is set in motion. Detailed formalization of this algorithm can be found in Appendix C.

**Theorem 2.** *Consider a feature index set* $\mathcal{A} = \{1, 2, ..., m\}$ *and a characteristic function* $v : 2^{\mathcal{A}} \to \mathbb{R}$ *such that* $v(\emptyset) = 0$. *Let* $\phi_j(v)$ *denote the importance of the* $j$*-th feature on* $v$ *such that* $\sum_{j=1}^m \phi(j) = v(\mathcal{A})$. *Assume that the indices in* $\mathcal{A}$ *are randomly distributed to* $K$ *parties with probabilities* $r_1, ..., r_K \sim Dir(\alpha_1, \ldots, \alpha_K)$. *Let* $Z_i$ *be the sum of feature importance for party* $i$. *Then, we have* $\forall i \in [1, K]$ *and* $\mathbb{E}[Z_i] \propto \alpha_i$.

The proof of Theorem 2 can be found in Appendix A, resembling the Dirichlet-multinomial mean proof but focusing on sum importance instead of feature counts. The metric of importance, $\phi_j(v)$, comprises the Shapley value and the recently proposed Shapley-CMI (Han et al., 2021). Theorem 2 asserts that the expected cumulative importance $\mathbb{E}[Z_i]$ of each party is proportional to the importance parameter $\alpha_i$. The Dirichlet-based split method ensures that: (1) a larger value of $\alpha_i$ leads to a higher expected value of $r_i$, thus a higher expected value of party importance, and (2) a smaller value of

$\|\{\alpha_i\}_{i=1}^K\|_2$ results in a larger variance in $r_i$, as well as more imbalanced importance among parties. Both properties are empirically validated in Appendix D.2. Hence, the proposed method naturally aligns with the requirements for feature importance. With $\alpha = 1$, Dirichlet-split mirrors a uniform distribution, incorporating random splits within the uniform scope. Even for manual equal splits lacking consistent criteria, a large $\alpha$ in Dirichlet-split can encapsulate them by yielding nearly equal feature distribution among parties.

## 3.2 SPLIT BY PARTY CORRELATION

This correlation-based feature-split algorithm (Alg. 1) is designed to allocate features across multiple parties based on a given correlation parameter $\beta$. The algorithm's operation is premised on a defined number of features for each party, represented as $m_1, \ldots, m_K$. Commencing with the initialization of a column permutation matrix $\mathbf{P}$ to an identity matrix (line 1), the algorithm proceeds to define a score function, $f(\mathbf{P}; \mathbf{X})$, which represents the overall correlation Icor after the features are permutated by $\mathbf{P}$ (line 2). Subsequently, the algorithm determines the range of the score function (lines 3-4). This forms the basis for calculating the target correlation $f^*(\mathbf{X}; \beta)$, which is a linear interpolation between the lower and upper bounds controlled by the correlation index $\beta$ (line 5). Next, the algorithm locates the optimal permutation matrix $\mathbf{P}^*$ by solving an permutation-based optimization problem. Notably, we employ the Biased Random-Key Genetic Algorithm (BRKGA) (Gonçalves and Resende, 2011) for this purpose. The final step of the algorithm splits the features according to the derived optimal permutation and the pre-set number of features for each party (lines 6-7).

---

**Algorithm 1:** Feature Splitting by Correlation

**Input:** Global dataset $\mathbf{X} \in \mathbb{R}^{n \times m}$, correlation index $\beta$, number of features $m_1, \ldots, m_K$
**Output:** Local dataasets $\mathbf{X}_1, \ldots, \mathbf{X}_K$
1  $\mathbf{P} \leftarrow \mathbf{I}$;                                    /* Initiate permutation matrix */
2  $f(\mathbf{P}; \mathbf{X}) := \mathrm{Icor}(\mathbf{X}_1^P, \ldots, \mathbf{X}_K^P) \ s.t. \ \mathbf{X}_1^P, \ldots, \mathbf{X}_K^P \leftarrow$ split features of $\mathbf{XP}$ by $m_1, \ldots, m_K$;
3  $f_{min}(\mathbf{X}) = \min_{\mathbf{P}} f(\mathbf{P}; \mathbf{X})$;                     /* Calculate lower bound */
4  $f_{max}(\mathbf{X}) = \max_{\mathbf{P}} f(\mathbf{P}; \mathbf{X})$;                     /* Calculate upper bound */
5  $f^*(\mathbf{X}; \beta) \leftarrow (1 - \beta) f_{min}(\mathbf{X}) + \beta f_{max}(\mathbf{X})$;   /* Calculate target correlation */
6  $\mathbf{P}^* \leftarrow \arg\min_{\mathbf{P}} |f(\mathbf{P}; \mathbf{X}) - f^*(\mathbf{X}; \beta)|$;     /* Find the permutation matrix */
7  $\mathbf{X}_1^P, \ldots, \mathbf{X}_K^P \leftarrow$ split features of $\mathbf{XP}^*$ by $m_1, \ldots, m_K$;
8  **return** $\mathbf{X}_1, \ldots, \mathbf{X}_K$

---

The efficiency of the optimization process, involving numerous Icor invocations, is crucial. For smaller datasets, Singular Value Decomposition (SVD) (Baker, 2005) is used for direct singular value computation. However, for high-dimensional datasets, we employ truncated SVD (Hansen, 1990) estimates the largest top-$d_t$ singular values, assuming the remainder as zero for standard variance calculation. The ablation study of $d_t$ is included in Appendix G.6. Our experiments, detailed in Appendix D.2, confirm the efficacy of both split methods.

## 3.3 COMPARE FEATURE SPLIT ACROSS GLOBAL DATASETS

The metrics presented in Section 2 facilitate meaningful comparisons of feature splits within the same global datasets but fall short when comparing across different datasets. To bridge this gap and enable a comparison between real and synthetic VFL datasets, we introduce methods to map these metrics to two values: $\alpha$ and $\beta$, where $\alpha$ indicates party balance and $\beta$ indicates party correlation. Consequently, this mapping enables a direct comparison between feature splits originating from real and synthetic VFL datasets, as demonstrated in Figure 1b.

To estimate $\alpha$, the importance of each party is calculated by Shapley values. These importance are then normalized and treated as Dirichlet parameters $\alpha_i$ for each party $P_i$, in line with Theorem 2. To approximate the scale of the Dirichlet parameters and align them with the generation of synthetic datasets, we find a symmetric Dirichlet distribution $\mathrm{Dir}(\alpha)$ that has the same variance as $\mathrm{Dir}(\alpha_1, \ldots, \alpha_K)$, as given in Proposition 4. This value of $\alpha$ reflects the variance of party importance. The proof is provided in Appendix A.

**Proposition 4.** *Given a Dirichlet distribution $Dir(\alpha_1, \ldots, \alpha_K)$ with mean variance $\sigma$, symmetric Dirichlet distribution $Dir(\alpha)$ that has the same mean variance $\sigma$ if $\alpha = \frac{K-1-K^2\sigma}{K^3\sigma}$.*

To estimate $\beta$, we start by computing the potential minimum and maximum values of Icor by shuffling the features among parties, denoted as $\text{Icor}_{\min}, \text{Icor}_{\max}$. Next, we estimate the Icor of the actual dataset, $\text{Icor}_{\text{real}}$, and derive the $\beta$ value using $\beta = \min\left\{\max\left\{\frac{\text{Icor}_{\text{real}} - \text{Icor}_{\min}}{\text{Icor}_{\max} - \text{Icor}_{\min}}, 0\right\}, 1\right\}$. It is important to note that in real-world scenarios, $\text{Icor}_{\text{real}}$ might fall slightly outside the range of $\text{Icor}_{\min}, \text{Icor}_{\max}$ due to the constraints of optimization algorithms. To rectify this, we clip the estimated $\beta$ to ensure $\beta \in [0, 1]$.

## 4  EXPERIMENT

This section benchmarks cutting-edge VFL algorithms, with a detailed review in Section 4.1. Experimental settings are outlined in Section 4.2, and results regarding VFL accuracy and synthetic-real correlation are in Sections 4.3 and 4.4, respectively. Further evaluations, such as real communication cost, scalability, training time, and real dataset performance, are in Appendix G. Each experiment elucidates results and provides relevant insights, highlighting (1) the performance-communication tradeoff of NN-based and boosting-based methods, (2) the performance similarity between synthetic and real VFL datasets under the same $\alpha, \beta$, and (3) the scalability potential of VFL algorithms.

### 4.1  REVIEW OF VFL ALGORITHMS

This section reviews existing VFL algorithms, with a focus on accuracy, efficiency, and communication cost. VertiBench concentrates on common supervised learning tasks such as classification and regression within synchronized parties, summarized in Table 1. Notably, this benchmark excludes studies exploring other aspects (Jin et al., 2021; Qi et al., 2022; Jiang et al., 2022) and other tasks (Chang et al., 2020; Li et al., 2021b; Chen and Zhang, 2022; He et al., 2022; Li et al., 2022b). Since most VFL algorithms presume exact inter-party data linking, we adopt this approach in VertiBench, despite recent contrary findings (Wu et al., 2022a; Nock et al., 2021) that this assumption may not be true. We refer to parties with and without labels as *primary* and *secondary parties* respectively.

Table 1: Summary of existing VFL algorithms

| Category | Model[1] | Algorithm | Contribution | Reference | Data[2] | Feature[3] |
|---|---|---|---|---|---|---|
| **Ensemble-based** | Any | AL | Accuracy | (Xian et al., 2020) | Syn | Manual |
| | | GAL | Accuracy | (Diao et al., 2022) | Syn | Manual |
| **Split-based** | NN | SplitNN | Accuracy | (Vepakomma et al., 2018) | Syn | N/A |
| | | C-VFL | Communication | (Castiglia et al., 2022) | Syn | Manual |
| | | BlindFL | Efficiency | (Fu et al., 2022b) | Syn | Manual |
| | | FedOnce | Communication | (Wu et al., 2022b) | Syn | Random |
| | GBDT | SecureBoost | Accuracy | (Cheng et al., 2021) | Syn | Manual |
| | | Pivot | Accuracy | (Wu et al., 2020) | Syn | Manual |
| | | FedTree | Accuracy, Efficiency | (Li et al., 2023) | Syn | Random |
| | | VF2Boost | Efficiency | (Fu et al., 2021) | Syn | Manual |
| | RF | Fed-Forest | Communication | (Liu et al., 2020) | Syn | Random |

[1] Abbreviations: NN - neural network; GBDT - gradient boosting decision trees; RF - random forest; Any - model-agnostic.
[2] Dataset in experiments: Syn - synthetic datasets partitioned from global datasets.
[3] Datasets used in the experiments: Manual - features manually split without specific reasons; Random - features randomly split without explanation; N/A - no VFL experiments conducted.

Most of the existing VFL methods can be categorized into *ensemble-based* and *split-based*. Ensemble-based methods have each party maintain a full model for local prediction and use collaborative ensemble techniques during training. Conversely, split-based methods delegate each party with a portion of the model, representing different inference stages. A comprehensive comparison is in Appendix B. In this paper, we concentrate on the primary types of VFL, acknowledging that there are various subtypes as identified in (Liu et al., 2022). Exploring these subtypes in depth will be an objective of our future research efforts.

In our experiments, we evaluate various VFL algorithms, including split-NN-based (e.g., SplitNN, C-VFL, FedOnce), split-GBDT-based (FedTree), and ensemble-based (GAL). For fairness, evaluations exclude encryption or noise. Noting minor variances among split-GBDT-based methods such as FedTree and SecureBoost, FedTree is used as a representative in our experiments.

## 4.2 EXPERIMENTAL SETTINGS

This subsection includes the datasets and training method. Detailed dataset specifications, environments, and hyperparameter settings can be found in Appendix F.

**Datasets.** Our experiments utilize 11 datasets: nine centralized ones (`covtype` (Blackard, 1998), `msd` (Bertin-Mahieux, 2011), `gisette` (Guyon et al., 2008), `realsim` (Andrew, 2015), `epsilon` (Guo-Xun et al., 2008), `letter` (Slate, 1991), `radar` (Khosravi, 2020), `MNIST` (Deng, 2012), `CIFAR10` (Krizhevsky and Hinton, 2009)), and two real-world VFL datasets (`NUS-WIDE` (Chua et al., 2009), `Vehicle` (Duarte and Hu, 2004)), with detailed descriptions available in Appendix F. The `msd` dataset is used for regression tasks, while the others cater to classification tasks. Each dataset is partitioned into 80% training and 20% testing instances except `NUS-WIDE`, `MNIST`, and `CIFAR10` with pre-defined test set. The datasets' features are distributed among multiple parties (typically four), split based on party importance ($\alpha$) or correlation ($\beta$). In the correlation-based split, each party is assigned an equal number of features.

**Training.** For classification tasks, we use accuracy as the evaluation metric, while regression tasks are evaluated using the Root Mean Square Error (RMSE). To ensure the reliability of our results, we conduct five runs for each algorithm, using seeds ranging from 0 to 4 to randomly split the datasets for each run, and then compute their mean metrics and standard deviation. Detailed hyper-parameter settings for each algorithms are provided in Appendix F.

## 4.3 VFL ACCURACY

In this subsection, we assess the impact on the performance of VFL algorithms when varying $\alpha$ and $\beta$. Our analysis includes all the three VFL categories in Table 1. The performance is summarized in Figure 3 and detailed in Table 9 in Appendix G. The result on `msd` dataset provides similar insights to others, thus only included in Table 9. From our exploration, we can draw three key observations.

**Split parameters $\alpha$ and $\beta$ significantly affect VFL algorithm performance, depending on the algorithm and dataset.** SplitNN and FedTree show stable performance across various $\alpha$ and $\beta$ settings. In contrast, C-VFL demonstrates notable performance fluctuations: up to 10% on `epsilon` and 40% on `letter` with varying $\alpha$. GAL performs better on imbalanced datasets (affected by $\alpha$ by 8% on `letter` and `radar`, 2-5% on others) and is minimally influenced by $\beta$. FedOnce, favoring balanced and highly correlated datasets, is affected by $\alpha$ (5-10% on `letter`, `gisette`, `epsilon`) and by $\beta$ (1-3% on `covtype`, `epsilon`). These findings highlight the need for comprehensive evaluations across a range of $\alpha$ and $\beta$ to determine VFL algorithms' robustness.

**SplitNN often leads in accuracy across most datasets; however, the performance of split-GBDT-based and ensemble-based methods can vary significantly depending on the dataset.** As anticipated, given its iterative transmission of substantial representations and gradients, SplitNN often outperforms other methods across a majority of datasets. Comparatively, the performance of FedTree and GAL is dataset-dependent. FedTree is well-suited to high-dimensional, smaller datasets like `gisette`, but struggles with larger datasets like `epsilon` and `covtype`. GAL, on the other hand, performs admirably with binary classification and regression tasks, though its performance drops significantly as the number of classes increases, as observed on the `covtype` and `letter` dataset.

**The compression of SplitNN renders them particularly affected by party imbalance.** C-VFL, modelled after SplitNN, exhibits the least accuracy among tested baselines due to its compression approach. Moreover, C-VFL exhibits marked sensitivity to the imbalance level, $\alpha$. Specifically, at $\alpha = 0.1$, its accuracy on datasets like `letter` and `epsilon` scarcely surpasses random guessing. However, C-VFL thrives in highly imbalanced split of `radar` dataset. This data-dependent behavior underscores an urgent need to refine compression techniques for VFL tailored to varying imbalances.

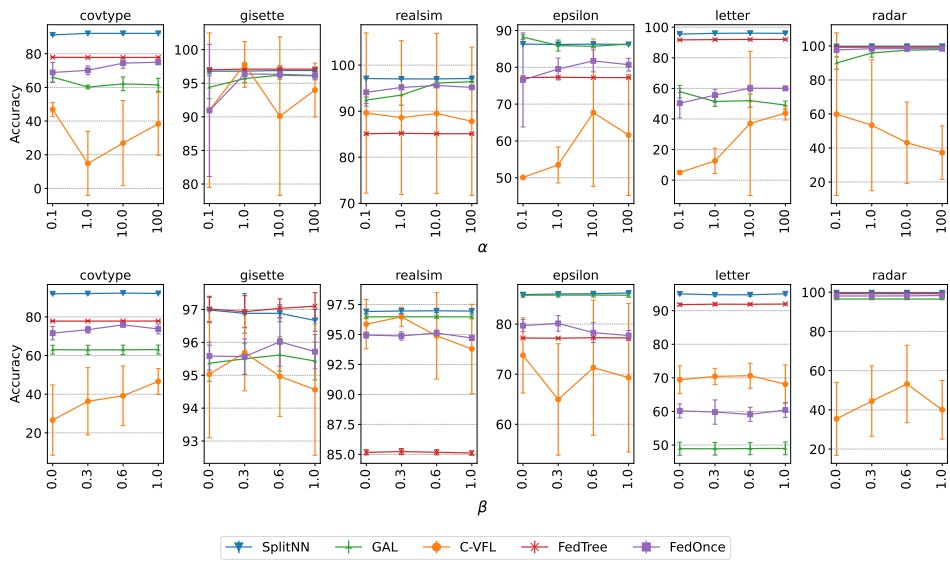

Figure 3: Accuracy of VFL algorithms on different datasets varying imbalance and correlation

### 4.4 PERFORMANCE CORRELATION: VERTIBENCH SCOPE VS. REAL SCOPE

In assessing the performance correlation between VertiBench-synthetic and real VFL datasets, we use derived $\alpha$ and $\beta$ values of `NUS-WIDE` and `Vehicle` (Section 3.3) to generate comparable synthetic datasets. To evaluate the relative performance of each algorithm, we calculate the accuracy differences between `Vehicle`-synthetic and `NUS-WIDE`-synthetic datasets for each algorithm and compare with real dataset accuracy differences, with further details in Appendix G.8.

Our experiment reveals a positive correlation between relative algorithm performance on synthetic datasets with matching $\alpha$ and $\beta$, and their performance on real VFL datasets. This indicates that, under the same $\alpha$ or $\beta$, higher mean accuracy on synthetic datasets typically implies better performance on real VFL datasets, thus affirming the relevance of VertiBench-synthetic datasets in approximating real VFL performance.

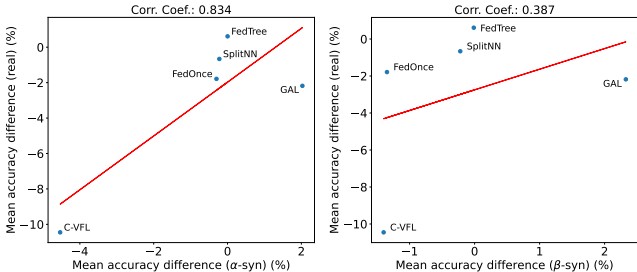

Figure 4: Mean accuracy differences: synthetic datasets vs. real datasets

## 5 CONCLUSION

We introduce VertiBench, a refined benchmarking tool for Vertical Federated Learning (VFL), adept at generating a variety of synthetic VFL datasets from a single global dataset. The scope of VertiBench extends beyond the confines of existing uniform and real scopes, shedding light on VFL scenarios previously unexplored. Our findings underscore performance variations under diverse data partitions, emphasizing the need to evaluate VFL algorithms across varied feature splits for enhanced insights into their real-world applicability.

## 6 REPRODUCIBILITY STATEMENT

The code for this study is accessible via a GitHub repository (Wu et al., 2023a), accompanied by a `README.md` file that provides guidelines for environment setup and result reproduction. Comprehensive proofs of all theoretical results are meticulously detailed in Appendix A. Further, Appendix F offers a detailed description of dataset specifications and hyperparameter configurations.

## ACKNOWLEDGEMENT

This research is supported by the National Research Foundation Singapore and DSO National Laboratories under the AI Singapore Programme (AISG Award No: AISG2-RP-2020-018). Any opinions, findings and conclusions or recommendations expressed in this material are those of the authors and do not reflect the views of National Research Foundation, Singapore. This work is supported in part by AMD under the Heterogeneous Accelerated Compute Clusters (HACC) program.

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

# Appendix

## Table of Contents

## A PROOF

**Proposition 1.** *The probability mass function can be written as*

$$\log \mathbb{P}(y|X_K, ..., X_1) = \sum_{i=1}^{K} \log \frac{\mathbb{P}(y|X_k, ..., X_1)}{\mathbb{P}(y|X_{k-1}, ..., X_1)} + \log \mathbb{P}(y) \tag{4}$$

*Proof.* According to the definition of conditional probability, this marginal distribution can be written as

$$
\mathbb{P}(y|X_K, ..., X_1)
$$

$$
= \frac{\mathbb{P}(y, X_K, ..., X_1)}{\mathbb{P}(X_K, ..., X_1)}
$$

$$
= \frac{\mathbb{P}(y)\mathbb{P}(X_1|y)\prod_{k=2}^{K}\mathbb{P}(X_k|y, X_{k-1}, ..., X_1)}{\mathbb{P}(X_1)\prod_{k=2}^{K}\mathbb{P}(X_k|X_{k-1}, ..., X_1)} \tag{5}
$$

$$
= \mathbb{P}(y)\frac{\mathbb{P}(X_1|y)}{\mathbb{P}(X_1)}\prod_{k=2}^{K}\frac{\mathbb{P}(X_k|y, X_{k-1}, ..., X_1)}{\mathbb{P}(X_k|X_{k-1}, ..., X_1)}
$$

Denoting

$$
c_k = \log\frac{\mathbb{P}(X_k|y, X_{k-1}, ..., X_1)}{\mathbb{P}(X_k|X_{k-1}, ..., X_1)}, \quad c_1 = \log\frac{\mathbb{P}(X_1|y)}{\mathbb{P}(X_1)} \tag{6}
$$

Adding logarithm on both sides, we have

$$
\log\mathcal{P}(y|X_K, ..., X_1) = \sum_{i=1}^{K}\log c_i + \log\mathbb{P}(y) \tag{7}
$$

Furthermore, we have

$$
\begin{aligned}
c_k &= \frac{\mathbb{P}(X_k|y, X_{k-1}, ..., X_1)}{\mathbb{P}(X_k|X_{k-1}, ..., X_1)} \\
&= \frac{\mathbb{P}(X_k, y|X_{k-1}, ..., X_1)}{\mathbb{P}(X_k|X_{k-1}, ..., X_1)\mathbb{P}(y|X_{k-1}, ..., X)} \\
&= \frac{\mathbb{P}(y|X_k, ..., X_1)}{\mathbb{P}(y|X_{k-1}, ..., X_1)}
\end{aligned} \tag{8}
$$

Combining (6) and (8), we have

$$
\log\mathcal{P}(y|X_K, ..., X_1) = \sum_{i=1}^{K}\log\frac{\mathbb{P}(y|X_k, ..., X_1)}{\mathbb{P}(y|X_{k-1}, ..., X_1)} + \log\mathbb{P}(y) \tag{9}
$$

$\square$

**Proposition 2.** *For any real matrix* $\mathbf{X}$, *Pcor*$(\mathbf{X}, \mathbf{X}) = mcor(\mathbf{X}, \mathbf{X})$

*Proof.* In the context of the correlation matrix of $\mathbf{X}$, Pcor measures the standard deviation of singular values, while mcor (Taylor, 2020) measures the standard deviation of eigenvalues. Given that the correlation matrix is symmetric positive semi-definite, each eigenvalue corresponds to a singular value, resulting in Pcor equating to mcor. $\square$

**Proposition 3.** *For any two real matrices* $\mathbf{X}_1$ *and* $\mathbf{X}_2$, *Pcor*$(\mathbf{X}_1, \mathbf{X}_2) \in [0, 1]$

*Proof.* Denote the correlation matrix between $\mathbf{X}_1$ and $\mathbf{X}_2$ as $C$, the singular values of $C$ as $\Sigma = [\sigma_1, \ldots, \sigma_d]$. Then

$$
\text{Pcor}(\mathbf{X}_1, \mathbf{X}_2) = \sqrt{\frac{1}{d}\text{Var}(\Sigma)} = \sqrt{\frac{1}{d}(\mathbb{E}(\Sigma^2) - \mathbb{E}^2(\Sigma))} \tag{10}
$$

Thus,

$$
0 \le \text{Pcor}(\mathbf{X}_1, \mathbf{X}_2) \le \sqrt{\frac{1}{d}\mathbb{E}(\Sigma^2)} \tag{11}
$$

Given that every element in the correlation matrix is in range $[-1, 1]$

$$\mathbb{E}(\Sigma^2) = \text{tr}(C^T C)/d \leq d^2/d \leq d \tag{12}$$

Thus, $\text{Pcor}(\mathbf{X}_1, \mathbf{X}_2) \in [0, 1]$ $\qquad\qquad\qquad\square$

**Theorem 1.** *Consider a global dataset $\mathbf{X}$ comprising two independent datasets $\mathbf{D}_1, \mathbf{D}_2 \in \mathbb{R}^{n \times m}$, each of the same dimension. Independence implies that for any feature $a_i^{(1)}$ from $\mathbf{D}_1$ and any feature $a_j^{(2)}$ from $\mathbf{D}_2$, where $i, j \in [1, m]$, the correlation $Cor\left(a_i^{(1)}, a_j^{(2)}\right) = 0$. Furthermore, assume within $\mathbf{D}_1$ and $\mathbf{D}_2$, all features are perfectly correlated, such that for all pairs of distinct features $a_i^{(1)}, a_j^{(1)}$ in $\mathbf{D}_1$ and $a_i^{(2)}, a_j^{(2)}$ in $\mathbf{D}_2$, with $i, j \in [1, m]$ and $i \neq j$, the correlations satisfy $Cor\left(a_i^{(1)}, a_j^{(1)}\right) = 1$ and $Cor\left(a_i^{(2)}, a_j^{(2)}\right) = 1$ respectively. When the features of $\mathbf{X}$ are divided equally into two subsets, $\mathbf{X}_1$ and $\mathbf{X}_2$, such that each subset contains $m$ features, the overall inter-party correlation $Icor(\mathbf{X}_1, \mathbf{X}_2)$ satisfies*

$$Icor(\mathbf{X}_1, \mathbf{X}_2) \in \left[ -\frac{m}{\sqrt{m(m-1)}}, 0 \right].$$

*The lower bound occurs if and only if $\mathbf{X}_1$ comprises all features of either $\mathbf{D}_1$ or $\mathbf{D}_2$, with $\mathbf{X}_2$ containing the remaining features. The upper bound occurs if and only if $\mathbf{X}_1$ holds $m$ features from both $\mathbf{D}_1$ and $\mathbf{D}_2$, with $\mathbf{X}_2$ holding the remaining $m$ features from $\mathbf{D}_1$ and $\mathbf{D}_2$.*

*Proof.* Consider two sets of features, $\mathbf{X}_1$ and $\mathbf{X}_2$, derived from datasets $\mathbf{D}_1$ and $\mathbf{D}_2$. Let $\mathbf{X}_1$ contain $u$ features from $\mathbf{D}_1$ and $v$ features from $\mathbf{D}_2$, where $u, v \in [0, m]$ and $u + v = m$. The set $\mathbf{X}_2$ comprises the remaining features, specifically $v$ features from $\mathbf{D}_1$ and $u$ features from $\mathbf{D}_2$. The inner-party correlation matrices $\mathbf{C}_{11}$ for $\mathbf{X}_1$ and $\mathbf{C}_{22}$ for $\mathbf{X}_2$ are given by:

$$\mathbf{C}_{11} = \begin{bmatrix} \mathbb{1}^{u \times u} & \mathbb{0}^{u \times v} \\ \mathbb{0}^{v \times u} & \mathbb{1}^{v \times v} \end{bmatrix}, \mathbf{C}_{22} = \begin{bmatrix} \mathbb{1}^{v \times v} & \mathbb{0}^{v \times u} \\ \mathbb{0}^{u \times v} & \mathbb{1}^{u \times u} \end{bmatrix} \tag{13}$$

To calculate the $\text{Pcor}(\mathbf{X}_1, \mathbf{X}_1)$ and $\text{Pcor}(\mathbf{X}_2, \mathbf{X}_2)$, we need to determine the singular values of $\mathbf{C}_{11}$ and $\mathbf{C}_{22}$. As these are symmetric matrices, their singular values are the absolute values of their eigenvalues. We will illustrate the process using $\mathbf{C}_{11}$, with $\mathbf{C}_{22}$ following a similar pattern.

The characteristic equation of $\mathbf{C}_{11}$ is

$$\begin{vmatrix} \mathbf{U}_{11}^{u \times u} & \mathbb{0}^{u \times v} \\ \mathbb{0}^{v \times u} & \mathbf{V}_{11}^{v \times v} \end{vmatrix} = 0 \tag{14}$$

where $\mathbf{U}_{11}^{u \times u}$ and $\mathbf{V}_{11}^{v \times v}$ are matrices defined as

$$\mathbf{U}_{11}^{u \times u} = \mathbf{V}_{11}^{v \times v} = \begin{bmatrix} 1 - \lambda & 1 & \cdots & 1 \\ 1 & 1 - \lambda & \cdots & 1 \\ \vdots & \vdots & \ddots & \vdots \\ 1 & 1 & \cdots & 1 - \lambda \end{bmatrix}$$

Equation 14 simplifies to $\det(\mathbf{U}_{11}^{u \times u})\det(\mathbf{V}_{11}^{v \times v}) = 0$. The roots of $\det(\mathbf{U}_{11}^{u \times u}) = 0$ and $\det(\mathbf{V}_{11}^{v \times v}) = 0$ are the eigenvalues of $\mathbb{1}^{u \times u}$ and $\mathbb{1}^{v \times v}$ respectively. Since $\mathbb{1}^{u \times u}$ and $\mathbb{1}^{v \times v}$ are rank-1 matrices, they each have only one non-zero eigenvalue, $u$ and $v$ respectively. Therefore, $\mathbf{C}_{11}$ has two non-zero eigenvalues: $u$ and $v$. The Pcor for $\mathbf{X}_1$ is

$$\text{Pcor}(\mathbf{X}_1, \mathbf{X}_1) = \frac{1}{\sqrt{m}} \sqrt{\frac{u^2 + v^2}{m - 1}} \tag{15}$$

A similar calculation for $\mathbf{C}_{22}$ yields:

$$\text{Pcor}(\mathbf{X}_2, \mathbf{X}_2) = \frac{1}{\sqrt{m}} \sqrt{\frac{u^2 + v^2}{m - 1}} \tag{16}$$

Next, we consider the inter-party correlation matrices $\mathbf{C}_{12}$ and $\mathbf{C}_{21}$

$$\mathbf{C}_{12} = \begin{bmatrix} \mathbb{1}^{u \times v} & \mathbb{0}^{u \times u} \\ \mathbb{0}^{v \times v} & \mathbb{1}^{v \times u} \end{bmatrix}, \mathbf{C}_{21} = \mathbf{C}_{12}^T = \begin{bmatrix} \mathbb{1}^{v \times u} & \mathbb{0}^{v \times v} \\ \mathbb{0}^{u \times u} & \mathbb{1}^{u \times v} \end{bmatrix} \tag{17}$$

Since $\mathbf{C}_{21} = \mathbf{C}_{12}^T$, they share the same singular values. A key principle in linear algebra states that the singular values of a matrix remain invariant under rotation. This invariance arises because rotating a matrix is mathematically equivalent to multiplying it by an orthogonal matrix, a process that does not alter its singular values. By rotating $\mathbf{C}_{12}$ and $\mathbf{C}_{21}$ 90 degrees counter-clockwise, we obtain symmetric matrices $\mathbf{C}_{12}'$ and $\mathbf{C}_{21}'$, whose singular values are the absolute values of their eigenvalues. As $\mathbf{C}_{12}'$ is a rank-2 matrix, it has two non-zero eigenvalues.

The characteristic equation of $\mathbf{C}_{12}'$ is

$$\begin{vmatrix} \mathbf{U}_{12}^{u \times u} & \mathbb{1}^{u \times v} \\ \mathbb{1}^{v \times u} & \mathbf{V}_{12}^{v \times v} \end{vmatrix} = 0 \tag{18}$$

where $\mathbf{U}_{12}^{u \times u}$ and $\mathbf{V}_{12}^{v \times v}$ are defined as

$$\mathbf{U}_{12}^{u \times u} = \begin{bmatrix} -\lambda & 0 & \cdots & 0 \\ 0 & -\lambda & \cdots & 0 \\ \vdots & \vdots & \ddots & \vdots \\ 0 & 0 & \cdots & -\lambda \end{bmatrix}^{u \times u}, \mathbf{V}_{12}^{v \times v} = \begin{bmatrix} -\lambda & 0 & \cdots & 0 \\ 0 & -\lambda & \cdots & 0 \\ \vdots & \vdots & \ddots & \vdots \\ 0 & 0 & \cdots & -\lambda \end{bmatrix}^{v \times v} \tag{19}$$

Since we focused on non-zero eigenvalues ($\lambda \neq 0$), $\mathbf{U}_{12}^{u \times u}$ is invertible. According to the property of block matrices, Eq. 18 is equivalent to

$$\det(\mathbf{U}_{12}^{u \times u}) \det(\mathbf{V}_{12}^{v \times v} - \mathbb{1}^{v \times u}(\mathbf{U}_{12}^{-1})^{u \times u}\mathbb{1}^{u \times v}) = 0 \tag{20}$$

Since $\det(\mathbf{U}_{12}^{u \times u}) = (-\lambda)^u$, and

$$\det(\mathbf{U}_{12}^{-1}) = \begin{vmatrix} -\frac{1}{\lambda} & 0 & \cdots & 0 \\ 0 & -\frac{1}{\lambda} & \cdots & 0 \\ \vdots & \vdots & \ddots & \vdots \\ 0 & 0 & \cdots & -\frac{1}{\lambda} \end{vmatrix} = \left(-\frac{1}{\lambda}\right)^u \tag{21}$$

Given $\lambda \neq 0$, Eq. 20 is equivalent to

$$\begin{vmatrix} \frac{u}{\lambda} - \lambda & \frac{u}{\lambda} & \cdots & \frac{u}{\lambda} \\ \frac{u}{\lambda} & \frac{u}{\lambda} - \lambda & \cdots & \frac{u}{\lambda} \\ \vdots & \vdots & \ddots & \vdots \\ \frac{u}{\lambda} & \frac{u}{\lambda} & \cdots & \frac{u}{\lambda} - \lambda \end{vmatrix}^{v \times v} = 0 \tag{22}$$

$$\begin{vmatrix} u - \lambda^2 & u & \cdots & u \\ u & u - \lambda^2 & \cdots & u \\ \vdots & \vdots & \ddots & \vdots \\ u & u & \cdots & u - \lambda^2 \end{vmatrix}^{v \times v} = 0 \tag{23}$$

The roots of Eq. 23 are the eigenvalues of $u \cdot \mathbb{1}^{v \times v}$, which is a rank-1 matrix. The only non-zero eigenvalue of $u \cdot \mathbb{1}^{v \times v}$ is $uv$, leading to $\lambda^2 = uv$. Thus, the two non-zero eigenvalues of $\mathbf{C}_{12}'$ are $\pm\sqrt{uv}$, and the Pcor of $\mathbf{C}_{12}$ is

$$\mathrm{Pcor}(\mathbf{X}_1, \mathbf{X}_2) = \frac{1}{\sqrt{m}}\sqrt{\frac{2uv}{m-1}} \tag{24}$$

Combining Eq. 15, 16, and 24, the overall inter-party correlation (Icor) is derived as

$$\begin{aligned} \mathrm{Icor}(\mathbf{X}_1, \mathbf{X}_2) &= \frac{1}{2}\left(2 * \mathrm{Pcor}(\mathbf{X}_1, \mathbf{X}_2) - \mathrm{Pcor}(\mathbf{X}_1, \mathbf{X}_1) - \mathrm{Pcor}(\mathbf{X}_2, \mathbf{X}_2)\right) \\ &= \frac{1}{\sqrt{m}}\sqrt{\frac{2uv}{m-1}} - \frac{1}{\sqrt{m}}\sqrt{\frac{u^2 + v^2}{m-1}} \\ &= \frac{1}{\sqrt{m(m-1)}}\left(\sqrt{2uv} - \sqrt{u^2 + v^2}\right) \end{aligned} \tag{25}$$

By the AM-HM inequality, $\text{Icor}(\mathbf{X}_1, \mathbf{X}_2) \leq 0$, with equality if and only if $u = v = \frac{m}{2}$.

For the lower bound, given $u = m - v$, expressing Icor in terms of $v$ gives

$$\text{Icor}(\mathbf{X}_1, \mathbf{X}_2) = \frac{1}{\sqrt{m(m-1)}} \left( \sqrt{2mv - 2v^2} - \sqrt{2v^2 - 2mv + m^2} \right) \tag{26}$$

Letting $t = 2mv - 2v^2 (v \in [0, m])$, it holds $t \in [0, m^2/2]$. Consider the function $f(t) = \sqrt{t} - \sqrt{m^2 - t}$, the derivative of $f(t)$ ensures

$$\begin{aligned} f'(t) &= \frac{1}{2\sqrt{t}} + \frac{1}{2\sqrt{m^2 - t}} \\ &= \frac{\sqrt{m^2 - t} - \sqrt{t}}{2\sqrt{t(m^2 - t)}} \geq 0 \end{aligned} \tag{27}$$

Therefore, $f(t)$ and $\text{Icor}(\mathbf{X}_1, \mathbf{X}_2)$ are monotonically non-descreasing w.r.t. $t$. The lower bound of Icor is reached when $t = 0$, yielding

$$\text{Icor}(\mathbf{X}_1, \mathbf{X}_2) \geq -\frac{m}{\sqrt{m(m-1)}} \tag{28}$$

This condition holds if and only if $v = 0$ or $v = m$. $\qquad\square$

**Theorem 2.** *Consider a feature index set $\mathcal{A} = \{1, 2, ..., m\}$ and a characteristic function $v : 2^{\mathcal{A}} \to \mathbb{R}$ such that $v(\emptyset) = 0$. Let $\phi_j(v)$ denote the importance of the $j$-th feature on $v$ such that $\sum_{j=1}^m \phi(j) = v(\mathcal{A})$. Assume that the indices in $\mathcal{A}$ are randomly distributed to $K$ parties with probabilities $r_1, ..., r_K \sim Dir(\alpha_1, \ldots, \alpha_K)$. Let $Z_i$ be the sum of feature importance for party $i$. Then, we have $\forall i \in [1, K]$ and $\mathbb{E}[Z_i] \propto \alpha_i$.*

*Proof.* For each feature $j$ assigned to party $i$ with probability $r_i$, we define the feature importance $Y_{ij}$ as:

$$Y_{ij} = \begin{cases} \phi_j(v), & w.p.\ r_i \\ 0, & w.p.\ 1 - r_i \end{cases} \tag{29}$$

By leveraging the property of linearity of expectation, we find that:

$$\mathbb{E}[Z_i] = \sum_{j=1}^m \mathbb{E}[Y_{ij}] = \sum_{j=1}^m \phi_j(v)\mathbb{E}[r_i] = \mathbb{E}[r_i] \sum_{j=1}^m \phi(j) \tag{30}$$

Given that $\sum_{j=1}^m \phi(j) = v(\mathcal{A})$, we derive:

$$\mathbb{E}[Z_i] = \mathbb{E}[r_i]v(\mathcal{A}) \tag{31}$$

Since the property of Dirichlet distribution asserts that $\alpha_i \approx \mathbb{E}[r_i]$, it holds

$$\mathbb{E}[Z_i] \propto \alpha_i v(\mathcal{A}) \tag{32}$$

Moreover, since $v(\mathcal{A})$ is a constant, it follows that:

$$\mathbb{E}[Z_i] \propto \alpha_i \tag{33}$$

$\qquad\square$

**Proposition 4.** *Given a Dirichlet distribution $Dir(\alpha_1, \ldots, \alpha_K)$ with mean variance $\sigma$, symmetric Dirichlet distribution $Dir(\alpha)$ that has the same mean variance $\sigma$ if $\alpha = \frac{K-1-K^2\sigma}{K^3\sigma}$.*

*Proof.* Suppose we have variables $X_1, \ldots, X_K$ following the Dirichlet distribution, denoted as $Dir(\alpha, \ldots, \alpha)$. Leveraging the inherent properties of the Dirichlet distribution, we can formulate the variance $\text{Var}(X_i)$ for all $i \in [1, K]$ as

$$\text{Var}(X_i) = \frac{K-1}{K^2(K\alpha + 1)} \tag{34}$$

The mean variance, denoted as $\sigma$, can subsequently be articulated in terms of the expected variance, $\mathbb{E}[\text{Var}(X_i)]$, as

$$\mathbb{E}[\text{Var}(X_i)] = \frac{K-1}{K^2(K\alpha+1)} = \sigma \tag{35}$$

Recognizing that for a Dirichlet distribution $\sigma > 0$ holds, we can transform the above equation to express $\alpha$ in terms of $\sigma$:

$$\alpha = \frac{K - 1 - K^2\sigma}{K^3\sigma} \tag{36}$$

$\square$

# B  DETAILS OF VFL ALGORITHMS

In this section, we provide a detailed comparison of the existing VFL algorithms as an extension of Section 4.1. This section critically reviews current VFL algorithms, with a focus on accuracy, efficiency, and communication size. VertiBench concentrates on standard supervised learning tasks such as classification and regression within synchronized parties, summarized in Table 1. Notably, this benchmark excludes studies exploring different VFL aspects such as privacy (Jin et al., 2021), fairness (Qi et al., 2022), data pricing (Jiang et al., 2022), asynchronization (Zhang et al., 2021b; Hu et al., 2019; Zhang et al., 2021a), latency (Fu et al., 2022a), and other tasks like unsupervised learning (Chang et al., 2020), matrix factorization (Li et al., 2021b), multi-task learning (Chen and Zhang, 2022), and coreset construction (Huang et al., 2022). While most VFL algorithms presume accurate inter-party data linking, we adopt this approach in VertiBench, despite recent contrary findings (Wu et al., 2022a; Nock et al., 2021) that this assumption may not be true.

The mainstream existing methods can be classified into two categories: *ensemble-based* and *split-based*. The distinguishing factor lies in the independent prediction capability of each party. Ensemble-based methods involve parties each maintaining a full model for local feature prediction, with collaborative ensemble methods during training, while split-based methods require each party to hold a partial model forming different inference stages of the full model. Consequently, split-based partial models cannot perform independent inference. For split-based models, our focus is on advanced models such as neural networks (NNs) and gradient boosting decision trees (GBDTs) (Chen and Guestrin, 2016), though VertiBench can accommodate various models (Hardy et al., 2017; Gu et al., 2020). Split-NN-based models are trained by transferring representations and gradients, while split-GBDT-models are trained by transferring gradients and histograms. While acknowledging the existence of various VFL algorithm subtypes (Li et al., 2022b), our current study primarily addresses the major types of VFL algorithms. The exploration of other subtypes remains a subject for future research.

## B.1  ENSEMBLE-BASED VFL ALGORITHMS

In the ensemble-based VFL algorithms detailed in Algorithm 2, each iteration $t$ commences with the primary party $P_1$ calculating the residual of the prior global model $F^{t-1}(\cdot)$ (line 3). This is followed by the communication of residuals $\mathbf{r}_1^t$ to the secondary parties $(P_2, \ldots, P_K)$. In the ensuing step (line 5), each secondary party trains a local model $f(\theta_i^t; \cdot)$ on its local data $\mathbf{X}_i$ to predict the residuals, subsequently sending the model parameters $\theta_i^t$ back to $P_1$. $P_1$ then aggregates these local models and updates the global model $F^t(\cdot)$ (line 6). This process iterates until a convergence criterion is achieved.

Specifics of residual sharing and model aggregation depend on algorithm design. In AL, residuals are shared among parties, and models are aggregated through summation. Conversely, in GAL, pseudo residuals (i.e., gradients) are shared, and models are aggregated through a weighted summation. Furthermore, the aggregation weight in GAL can updated during the training process.

**Algorithm 2:** Outline of ensemble-based VFL algorithms

| | |
|---|---|
| **Input** | : Number of iterations $T$; number of parties $K$; data and labels of primary party $\mathbf{X}_1, \mathbf{y}$, learning rate $\eta^t$; data of secondary parties $\mathbf{X}_2, \ldots, \mathbf{X}_K$ |
| **Output** | : Models on $K$ parties $f(\theta_1; \cdot), \ldots, f(\theta_K; \cdot)$ |
| **Algorithms** | : AL, GAL |

$$\texttt{Residual}(F^{t-1}(\mathbf{X}), \mathbf{y}) = \begin{cases} \mathcal{L}(F^{t-1}(\mathbf{X}), \mathbf{y}) & \text{, AL} \\ \frac{\partial}{\partial F^{t-1}(\mathbf{X})} \mathcal{L}(F^{t-1}(\mathbf{X}), \mathbf{y}) & \text{, GAL} \end{cases}$$

$$\texttt{Merge}_{i=1}^{k} f(\theta_1^t; \mathbf{X}_1) = \begin{cases} \eta^t \sum_{i=1}^{k} f(\theta_i^t; \mathbf{X}_i) & \text{, AL} \\ \eta^t \sum_{i=1}^{k} w_i^t f(\theta_i^t; \mathbf{X}_i) & \text{, GAL} \end{cases}$$

1   Initialize $\theta_1^0, \ldots, \theta_K^0$; initialize $F^0 \leftarrow 0$;
2   **for** $t = 1, \ldots, T$ **do**
3     $\mathbf{r}_1^t \leftarrow \texttt{Residual}(F^{t-1}(\mathbf{X}), \mathbf{y})$;        /* $P_1$: Compute residual */
4     **for** $i = 1, \ldots, k$ **do**
5        $\theta_i^t \leftarrow \arg\min_{\theta_i^t} \mathcal{L}(f(\theta_i^t; \mathbf{X}_i), \mathbf{r}_1^t)$;     /* $P_K$: Optimize local model */
6     $F^t(\mathbf{X}) \leftarrow F^{t-1}(\mathbf{X}) + \texttt{Merge}_{i=1}^{k} f(\theta_1^t; \mathbf{X}_1)$;   /* $P_1$ Update ensemble model */

## B.2   SPLIT-NN-BASED VFL ALGORITHMS

As described in Algorithm 3, each iteration $t$ starts with the parties $P_i$ conducting forward-propagation on their local data $\mathbf{X}_i$ to derive local representation $\mathbf{Z}_i^t$ (line 4). These representations are subsequently forwarded to the primary party $P_1$. Depending on the iteration, $P_1$ then merges the local representations (line 6), further derives a global prediction $\hat{\mathbf{y}}^t$ with an aggregated model (line 7), updates the aggregated model parameters $\theta_1^t$ (line 8), and broadcast the encoded aggregation model $\theta_i^t$ to all parties (line 9). The parties $P_i$ then employ these encoded aggregation model to update their local models (line 11). This process is repeated until a specified criterion for stopping is met.

The specific methods of encoding, determining aggregation frequency, and merging are dependent on the algorithm design. In the process of forward-pass encoding, SplitNN sends local representations directly to Party $P_1$ for merging, while C-VFL compresses these representations before transmission. In contrast, BlindFL utilizes a source layer to encode local representations, ensuring privacy preservation. During backward-pass encoding, C-VFL transmits the top-k-compressed aggregation model. Both SplitNN and BlindFL initially compute the gradients with respect to $\mathbf{Z}$ and subsequently broadcast either the raw or source-layer encoded gradients to all the parties. Regarding aggregation frequency, C-VFL aggregates every $Q$ iterations to reduce communication cost, while both SplitNN and BlindFL aggregate at every iteration. For the merging process, SplitNN and C-VFL use concatenation of local representations, while BlindFL applies a secret-sharing summation with source-layer-encoded representations.

FedOnce (Wu et al., 2022b), a split-based VFL approach, distinguishes itself by its single-round communication protocol. Secondary parties initially engage in unsupervised learning, aiming to predict noise and subsequently develop local representations. These representations are then transferred to the primary party, which utilizes them in training a SplitNN model on the primary dataset. This method notably reduces both the communication size and the synchronization overhead.

## B.3   SPLIT-GBDT-BASED VFL ALGORITHMS

Outlined in Algorithm 4, each iteration $t$ initiates with the primary party $P_1$ encoding the gradient of residuals, yielding $\mathbf{r}^t$ (line 3). Following this, all parties $P_i$ calculate local histograms $\mathbf{H}_i^t$ utilizing their individual local data $\mathbf{X}_i$ and the encoded residuals $\mathbf{r}^t$ (line 5). These local histograms are then transmitted to $P_1$ for merging (line 6). In the next step, $P_1$ trains a decision tree using the merged histogram $\mathbf{H}^t$ and the encoded residuals $\mathbf{r}^t$ (line 7). The selected split points from this tree are communicated to the secondary party that possesses the split feature, which stores these split points for potential future requests during inference (line 9). Finally, $P_1$ updates the ensemble model $F^t$ with the newly trained tree (line 10). This sequence of operations continues until a set stopping condition is fulfilled.

**Algorithm 3:** Outline of split-NN-based VFL algorithms

| | |
|---|---|
| **Input** | :Number of iterations $T$; number of parties $K$; data and labels of primary party $\mathbf{X}_1, \mathbf{y}$, learning rate $\eta^t$; data of secondary parties $\mathbf{X}_2, \ldots, \mathbf{X}_K$; local epochs $Q$ |
| **Output** | :Models on $K$ parties $f(\theta_1; \cdot), \ldots, f(\theta_K; \cdot)$ |
| **Algorithms** | :SplitNN, C-VFL, BlindFL |

$$\texttt{Encode}(\mathbf{Z}) = \begin{cases} \mathbf{Z} & \text{, SplitNN} \\ \texttt{EmbeddingCompress}(\mathbf{Z}) & \text{, C-VFL} \\ \texttt{SourceLayer}(\mathbf{Z}) & \text{, BlindFL} \end{cases}$$

$$\texttt{Encode}(\theta_1^t) = \begin{cases} \frac{\partial}{\partial \mathbf{Z}_i^t} \mathcal{L}(\hat{\mathbf{y}}^t, \mathbf{y}; \theta_1^t) & \text{, SplitNN} \\ \texttt{TopK}(\theta_1^t) & \text{, C-VFL} \\ \texttt{SourceLayer}(\frac{\partial}{\partial \mathbf{Z}_i^t} \mathcal{L}(\hat{\mathbf{y}}^t, \mathbf{y}; \theta_1^t)) & \text{, BlindFL} \end{cases}$$

$$\texttt{condition}(t) = \begin{cases} \text{True} & \text{, SplitNN, BlindFL} \\ t \mod Q = 0 & \text{, C-VFL} \end{cases}$$

$$\texttt{Merge}_{i=1}^k \mathbf{Z}_i^t = \begin{cases} \texttt{Concatenate}(\mathbf{Z}^1, \ldots, \mathbf{Z}^t) & \text{, SplitNN, C-VFL} \\ \texttt{SecretSharingSum}(\mathbf{Z}^1, \ldots, \mathbf{Z}^t) & \text{, BlindFL} \end{cases}$$

1  Initialize $\theta_1^0, \ldots, \theta_K^0$;
2  **for** $t = 1, \ldots, T$ **do**
3      **for** $i = 1, \ldots, k$ **do**
4          $\mathbf{Z}_i^t \leftarrow \texttt{Encode}(f(\theta_i^t; \mathbf{X}_i))$;      `/* `$P_i$`:  Get local representations */`
5      **if** $\texttt{condition}(t)$ **then**
6          $\mathbf{Z}^t \leftarrow \texttt{Merge}_{i=1}^k \mathbf{Z}_i^t$;      `/* `$P_1$`:  Merge local representations */`
7          $\hat{\mathbf{y}}^t \leftarrow f(\theta_1^t; \mathbf{Z}^t)$;      `/* `$P_1$`:  Get global prediction */`
8          $\theta_1^t \leftarrow \theta_1^{t-1} - \eta^t \frac{\partial}{\partial \mathbf{Z}_1^t} \mathcal{L}(\hat{\mathbf{y}}^t, \mathbf{y})$;      `/* `$P_1$`:  Update aggregated model */`
9          $\mathbf{g}_i^t \leftarrow \texttt{Encode}(\theta_1^t)$;      `/* `$P_1$`:  Encode aggregated model */`
10     **for** $i = 1, \ldots, k$ **do**
11         $\theta_i^t \leftarrow \theta_i^{t-1} - \eta^t \mathbf{g}_i^t$;      `/* `$P_i$`:  Update local model */`

Depending on the algorithm, the specific techniques for encoding, computing histograms, merging, and updating modes differ. Pivot views the instance set in each node as confidential information, implementing homomorphic encryption on it. Conversely, SecureBoost, FedTree, and VF2Boost treat the instance set as public information and apply homomorphic encryption only on a specific set of instances. In terms of merging, SecureBoost and FedTree perform homomorphic decryption directly to acquire actual sum values after aggregating encrypted histograms. VF2Boost further introduces efficiency measures such as Polynomial-based histogram packing and reordered histogram accumulation, while Pivot utilizes multi-party computation supporting comparison to maintain the secrecy of values at all times. In terms of the update mode, SecureBoost, Pivot, and FedTree adopt a sequential approach, whereas VF2Boost utilizes a pipeline processing for speedup. Additionally, it is worth mentioning that SecureBoost employs a threshold for binary classification to enhance accuracy in the context of datasets with label imbalance.

## C  SPLIT METHOD DETAILS

In this section, we formally state out proposed importance-based feature-split algorithm in Algorithm 5. After initializing local datasets for each party (line 1), a series of probabilities $p_1, \ldots, p_K$ s.t. $\sum_{i=1}^K p_i = 1$ is sampled from a Dirichlet distribution, parameterized by $\alpha_1, \ldots, \alpha_K$ (line 2). For each feature, it then proceeds to randomly select a party $P_k$, according to the probabilities $p_k$, and assigns the respective feature to $P_k$ (lines 3-5). In order to address potential failures in algorithms when confronted with empty features, we can optionally initialize each party with a random feature prior to the commencement of the algorithm.

---

**Algorithm 4:** Outline of split-GBDT-based VFL algorithms

**Input**     : Number of iterations $T$; number of parties $K$; data and labels of primary party $\mathbf{X}_1, \mathbf{y}$, learning rate $\eta^t$; data of secondary parties $\mathbf{X}_2, \ldots, \mathbf{X}_K$
**Output**    : Models on $K$ parties $f(\theta_1; \cdot), \ldots, f(\theta_K; \cdot)$
**Algorithms**: SecureBoost, Pivot, FedTree, VF2Boost

$\mathtt{HE} = \mathtt{HomomorphicEncrypt}, \mathtt{HD} = \mathtt{HomomorphicDecrypt}, \mathtt{MPC} = \mathtt{MultiPartyComputationEncode}, \otimes$ is homomorphic multiplication, $\alpha$ is instance mask

$$\mathtt{Encode}(\mathbf{Z}) = \begin{cases} \mathtt{HE}(\mathbf{r} \cdot \alpha) & \text{, SecureBoost, FedTree, VF2Boost} \\ \mathbf{r} \otimes \mathtt{HE}(\alpha) & \text{, Pivot} \end{cases}$$

$$\mathtt{Hist}(\mathbf{X}, \mathbf{r}) = \begin{cases} \mathtt{HE}(\mathtt{Histogram}(\mathbf{X} \cdot \alpha, \mathbf{r})) & \text{, SecureBoost, FedTree, VF2Boost} \\ \mathtt{Histogram}(\mathbf{X}, \mathbf{r}) \otimes \mathtt{HE}(\alpha) & \text{, Pivot} \end{cases}$$

$$\mathtt{Merge}_{i=1}^k \mathbf{H}_i^t = \begin{cases} \mathtt{HD}(\mathtt{HESum}(\mathbf{H}_1^t, \ldots, \mathbf{H}_K^t)) & \text{, SecureBoost, FedTree} \\ \mathtt{HD}(\mathtt{Pack}(\mathtt{HESum}(\mathtt{Re\text{-}order}(\mathbf{H}_1^t, \ldots, \mathbf{H}_K^t)))) & \text{, VF2Boost} \\ \mathtt{MPC}(\mathtt{HESum}(\mathbf{H}_1^t, \ldots, \mathbf{H}_K^t)) & \text{, Pivot} \end{cases}$$

$$\mathtt{mode} = \begin{cases} \mathtt{sequence} & \text{, SecureBoost, Pivot, FedTree} \\ \mathtt{pipeline} & \text{, VF2Boost} \end{cases}$$

1  Initialize $\theta_1^0, \ldots, \theta_K^0$; let federated model $F^0(\cdot) \leftarrow 0$;
2  **for** $t = 1, \ldots, T$ **in** $\mathtt{mode}$ **do**
3  $\quad \mathbf{r}^t \leftarrow \mathtt{Encode}(\frac{\partial}{\partial F^{t-1}} \mathcal{L}(F^{t-1}(\mathbf{H}^{t-1}), \mathbf{y}));$  /* $P_1$: Encode gradients */
4  $\quad$ **for** $i = 1, \ldots, k$ **do**
5  $\quad\quad \mathbf{H}_i^t \leftarrow \mathtt{Hist}(\mathbf{X}_i, \mathbf{r}^t);$  /* $P_i$: Compute local histogram */
6  $\quad \mathbf{H}^t \leftarrow \mathtt{Merge}_{i=1}^k \mathbf{H}_i^t;$  /* $P_1$: Merge local histograms */
7  $\quad \theta_1^t \leftarrow \arg\min_{\theta_1^t} \mathcal{L}(f(\theta_1^t; \mathbf{H}^t), \mathbf{r}^t);$  /* $P_1$: Construct tree */
8  $\quad$ **for** $i = 2, \ldots, k$ **do**
9  $\quad\quad \theta_i^t \leftarrow$ Selected split points in $\mathbf{H}_i^t;$  /* $P_i$: Update secondary parties */
10 $\quad F^t(\mathbf{H}^t) \leftarrow F^{t-1}(\mathbf{H}^{t-1}) + f(\theta_1^t; \mathbf{H}^t);$  /* $P_1$: Update ensemble model */

---

---

**Algorithm 5:** Feature Splitting by Importance

**Input:** Global dataset $\mathbf{X} \in \mathbb{R}^{n \times m}$, importance factors $\alpha_1, \ldots, \alpha_K$
**Output:** Local dataasets $\mathbf{X}_1, \ldots, \mathbf{X}_K$
1  $\mathbf{X}_1, \ldots, \mathbf{X}_K \leftarrow \emptyset$;  /* Initialize local dataset for $P_i$ */
2  $p_1, \ldots, p_K \leftarrow \mathtt{Dirichlet}(\alpha_1, \ldots, \alpha_K)$; /* Sample probabilities for $K$ parties */
3  **for** $j = 1, \ldots, m$ **do**
4  $\quad k \leftarrow$ Random choice from $[1, K] \cap \mathbb{N}$ $w.p.$ $(p_1, \ldots, p_K)$;  /* Choose a party */
5  $\quad D_k \leftarrow D_k \cup \{\mathbf{X}[:, j]\}$;  /* Add feature $\mathbf{X}[:, j]$ to $P_k$ */
6  **return** $\mathbf{X}_1, \ldots, \mathbf{X}_K$

---

## D  EMPIRICAL VALIDATION OF SPLIT METHODS

To rigorously evaluate the practical performance of our proposed correlation evaluation metric and the correlation-based feature-split algorithm, we conduct a series of systematic experiments.

### D.1  CORRELATION EVALUATION METRIC

In order to validate the efficacy of the correlation evaluation metric, Pcor, we create two synthetic datasets, $\mathbf{X}_1$ ($m_1$ features) and $\mathbf{X}_2$ ($m_2$ features), using the $\mathtt{sklearn}$ library. Initially, party $P_1$ holds $\mathbf{X}_1$, and $P_2$ holds $\mathbf{X}_2$. Over the course of the experiment, we gradually transfer features from $\mathbf{X}_1$ to $P_2$, each time exchanging for a feature of $\mathbf{X}_2$. This process continues until all features of $\mathbf{X}_1$ end up on $P_2$, while the total number of features remains constant during the whole process.

Our observations, as presented in Figure 5a and 5b, reveal the following: (1) Pcor behaves similarly to mcor when evaluating inner-party correlation and shows a similar trend to mcor (Taylor, 2020) when assessing inter-party correlation. (2) Both Pcor and mcor exhibit the lowest inter-party correlation and the highest inner-party correlation at the extremities of the x-axis. This suggests that the datasets $\mathbf{X}_1$ and $\mathbf{X}_2$ are managed by distinct parties and exhibit complete independence. This pattern is also reflected in Figure 5c when Pcor is applied to parties of different dimensions. These observations validate the appropriateness of Pcor as a measure for evaluating inter-party correlation, even when the number of features between the two parties varies.

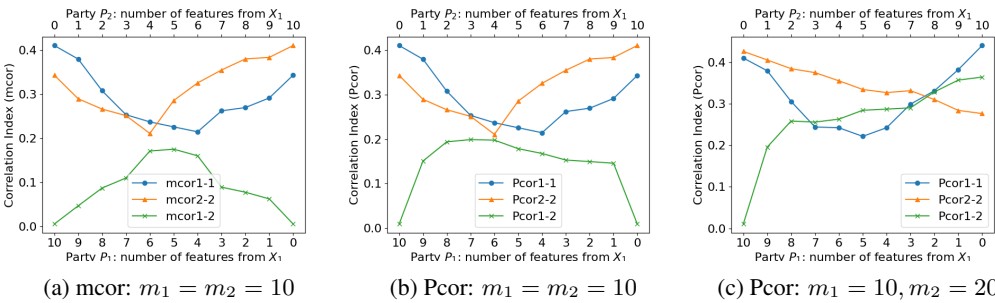

(a) mcor: $m_1 = m_2 = 10$     (b) Pcor: $m_1 = m_2 = 10$     (c) Pcor: $m_1 = 10, m_2 = 20$

Figure 5: The trend of correlation and metrics when exchanging features between two parties. mcor: multi-way correlation (Taylor, 2020); Pcor(i)-(j): Pcor($\mathbf{X}_i, \mathbf{X}_j$); mcor(i)-(j): mcor($\mathbf{X}_i, \mathbf{X}_j$).

### D.2 FEATURE-SPLIT ALGORITHM

**Importance-based Feature Split.** We rigorously investigate the correlation between $\{\alpha_i\}$ and party importance, using both Shapley value and Shapley-CMI for feature importance assessment. For property (1), in a two-party VFL experiment, we fix $\alpha_2 = 1$ and vary $\alpha_1$ from $10^{-2}$ to $10^3$. For each $\alpha_2$, we compute the scaled $\alpha_i$ as $\alpha_i / \sum_i \alpha_i$ and run 1,000 trials to estimate the party importance's expected value. The subsequent relationship between scaled $\alpha_i$ and the expected party importance is illustrated in Figure 6a and 6b, demonstrating clear proportionality.

For property (2), four parties are divided under a symmetric Dirichlet distribution (where $\forall i, \alpha_i = \alpha$) with $\alpha$ values ranging from $10^{-2}$ to $10^3$. For each $\alpha$, the standard variance of party importance is estimated over 1,000 trials, as visualized in Figure 6c and 6d. These plots signify a negative relationship between $\alpha = \|\{\alpha_i\}_{i=1}^K\|_2$ and the standard variance of party importance, extending the scenario of a random split leading to consistent party imbalance levels.

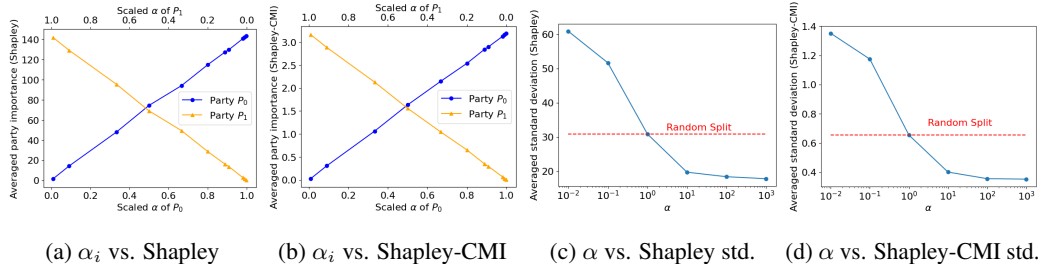

(a) $\alpha_i$ vs. Shapley    (b) $\alpha_i$ vs. Shapley-CMI    (c) $\alpha$ vs. Shapley std.    (d) $\alpha$ vs. Shapley-CMI std.

Figure 6: Relationship between $\alpha$ and party importance: (a)(b) Proportionality of scaled $\alpha_i$ to party $P_i$'s importance; (c)(d) Negative correlation between $\alpha$ and standard variance (std.) of party importance. The red dotted line indicates the importance variance of random split.

**Correlation-based Feature Split.** We turn our attention towards validating the efficacy of our proposed correlation-based feature-split algorithm. Three synthetic datasets, each encompassing 10 features, are independently generated using the `sklearn` library. These datasets are subsequently concatenated along the feature axis, yielding a global dataset with 30 features. This dataset, with features shuffled, is then split into three local datasets, each containing 10 features, deploying our proposed algorithm with $\beta$ values set at 0, 0.5, and 1.0.

Specifically, we split synthetic datasets with various $\beta$ values using Algorithm 1 (as shown in Figure 7a, 7b, and 7c) and compare this to a random split (Figure 7d). Our findings reveal that as $\beta$ increases, inter-party correlation correspondingly escalates. Additionally, as depicted in Figure 8, when we perform splits on a concatenated feature-shuffled real VFL dataset Vehicle ($\beta = 0$), **VertiBench, when parameterized with the same $\beta$, exactly reconstruct the real feature split of Vehicle dataset.** This indicates that VertiBench scope aligns well with the real scope which corresponds to a scenario with small inter-party correlation as further depicted in Figure 9. In contrast, a uniformly random split (uniform scope) produces a distinct correlation matrix.

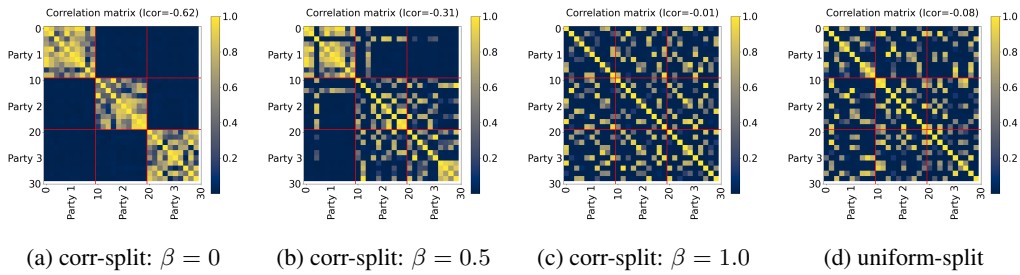

| (a) corr-split: $\beta = 0$ | (b) corr-split: $\beta = 0.5$ | (c) corr-split: $\beta = 1.0$ | (d) uniform-split |

Figure 7: Absolute correlation matrix of the global dataset with party boundaries indicated by red lines. Icor means inter-party correlation. (a),(b),(c) - correlation-based split; (d) uniform split.

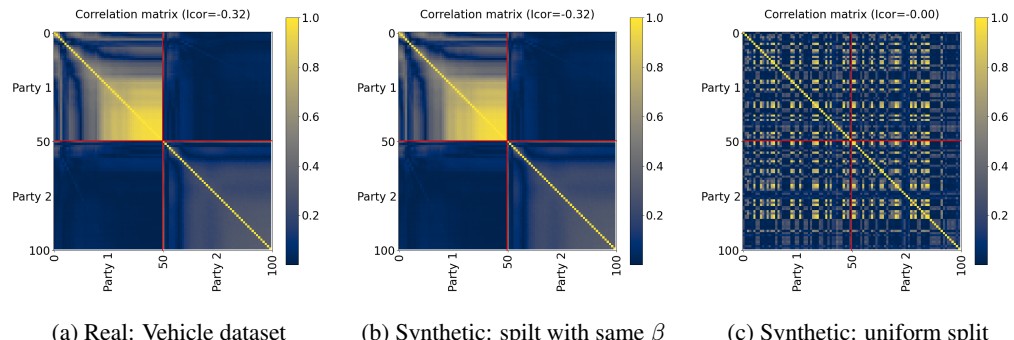

| (a) Real: Vehicle dataset | (b) Synthetic: spilt with same $\beta$ | (c) Synthetic: uniform split |

Figure 8: Absolute correlation matrix of real Vehicle dataset and synthetic datasets generated from feature-shuffled Vehicle - VertiBench with the same $\beta$ vs. uniform split

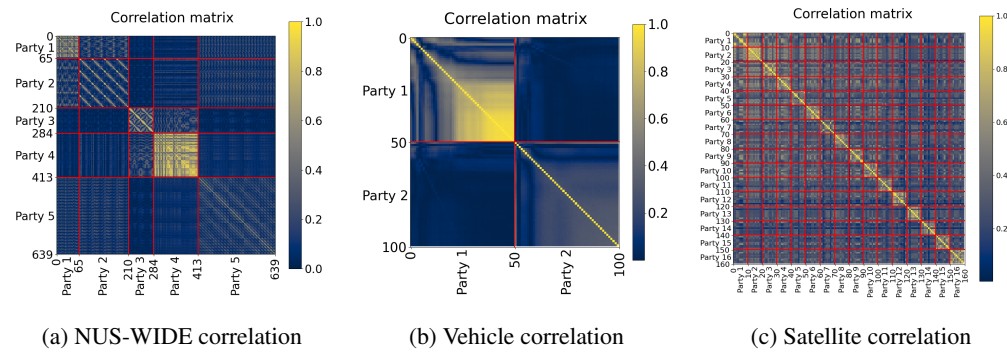

| (a) NUS-WIDE correlation | (b) Vehicle correlation | (c) Satellite correlation |

Figure 9: Absolute correlation matrix of real VFL datasets with party boundaries marked in red lines

### D.3 TIME EFFICIENCY OF SPLIT METHODS

Table 2 provides a summary of the estimated time requirements for our proposed split methods, with the I/O time for loading and saving datasets excluded. Notably, the importance-based split method demonstrates significant efficiency, typically completing within a minute. In contrast, the correlation-based split method requires a longer processing time, due to the need to resolve three

optimization problems. This time cost is especially pronounced on high-dimensional datasets, such as `realsim`, because the singular value decomposition (SVD) used in the correlation-based split algorithm is dependent on the number of features. Despite these differences in time consumption, both split methods prove capable of handling large datasets, accommodating instances up to 581k and features up to 20k, within a reasonable time frame.

Table 2: Estimated time cost of splitting methods (in seconds)

| Dataset | Performance of importance-based split | | | | Performance of correlation-based split | | | |
| --- | --- | --- | --- | --- | --- | --- | --- | --- |
| | $\alpha = 0.1$ | $\alpha = 1$ | $\alpha = 10$ | $\alpha = 100$ | $\beta = 0$ | $\beta = 0.3$ | $\beta = 0.6$ | $\beta = 1$ |
| covtype | 0.21 | 0.24 | 0.26 | 0.27 | 64.15 | 55.77 | 53.19 | 69.34 |
| msd | 0.29 | 0.27 | 0.29 | 0.30 | 85.39 | 68.88 | 64.76 | 62.01 |
| gisette | 0.23 | 0.29 | 0.25 | 0.27 | 6743.26 | 6388.56 | 5640.81 | 6733.69 |
| realsim | 30.11 | 29.80 | 30.60 | 26.68 | 13681.97 | 13381.56 | 11341.31 | 10057.42 |
| letter | 0.00 | 0.00 | 0.00 | 0.00 | 43.15 | 46.01 | 41.50 | 38.72 |
| epsilon | 12.67 | 12.30 | 10.36 | 12.28 | 4853.63 | 4395.76 | 4105.53 | 3621.99 |
| radar | 0.14 | 0.15 | 0.16 | 0.15 | 308.61 | 274.06 | 236.73 | 299.12 |

To further evaluate the performance, we specifically arranged experiments to evaluate the computational intensity of the correlation-based splitting method. We conducted tests on synthetic datasets of varying dimensions or sizes generated by `sklearn`. Using a Pearson-based splitting method on a single RTX 3090 GPU, the results are detailed in Figure 10.

Scalability experiments indicate the efficiency of the proposed split method. For the importance-based split method, it avoids the computational demands of Shapley value calculations while retaining related properties. This method completed within 30 seconds across all tested datasets, ensuring scalability for large datasets. The correlation-based method effectively handles datasets with up to 10k features, aligning with the dimensional needs of common VFL datasets such as `realsim` and `CIFAR10`. Meanwhile, the number of parties hardly affect the efficiency under a fixed global dataset.

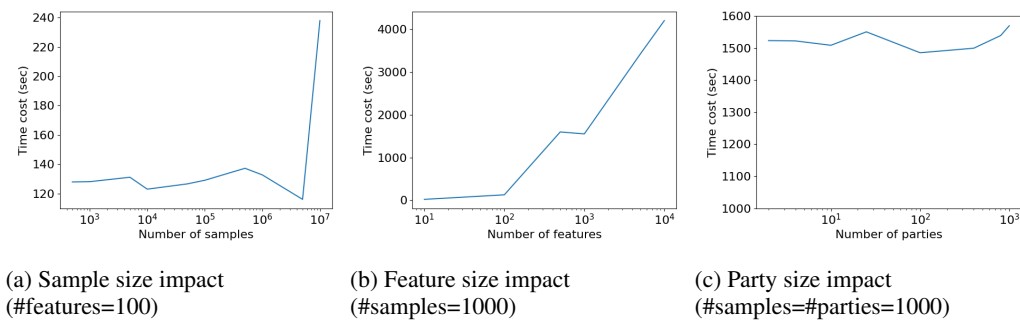

(a) Sample size impact (#features=100)

(b) Feature size impact (#samples=1000)

(c) Party size impact (#samples=#parties=1000)

Figure 10: Scalability of correlation-based split algorithm on #samples and #features

# E   REAL DATASET CONSTRUCTION

In this section, we outline the construction process of `Satellite`, which was adapted from the WorldStrat dataset (Cornebise et al., 2022), originally intended for high-resolution imagery analysis. The `Satellite` encompasses Point of Interest (POI) data, each associated with one or more Areas of Interest (AOI). Every AOI incorporates a unique location identifier, a land type, and 16 low-resolution, 13-channel images, each taken during a satellite visit to the location.

During the data cleaning phase, we scrutinize the dataset thoroughly, identifying and removing 67 incomplete data records that have an insufficient number of low-resolution images. Furthermore, given the inconsistent widths and heights of images across different locations, we standardize the size

of all images to a 158x158 square via bicubic interpolation. Additionally, the pixel values of each image are scaled to integer values within the range of $[0, 255]$.

`Satellite` forms a practical VFL scenario for location identification based on satellite imagery. Each AOI, with its unique location identifier, is captured by 16 satellite visits. Assuming each visit is carried out by a distinct satellite organization, these organizations aim to collectively train a model to classify the land type of the location without sharing original images. `Satellite` encompasses four land types as labels, namely `Amnesty POI` (4.8%), `ASMSpotter` (8.9%), `Landcover` (61.3%), and `UNHCR` (25.0%), making the task a 4-class classification problem of 3,927 locations.

As depicted in Figure 11, the consistency in capturing the same location 16 times presents variations in image quality. These discrepancies arise due to the changing weather and lighting conditions experienced during each satellite visit. Hence, each of these 16 satellite visits can equivalently be considered as 16 separate parties. All 13-channel images in the satellite dataset that correspond to the description of Sentinel-2 Bands as shown in Table 3.

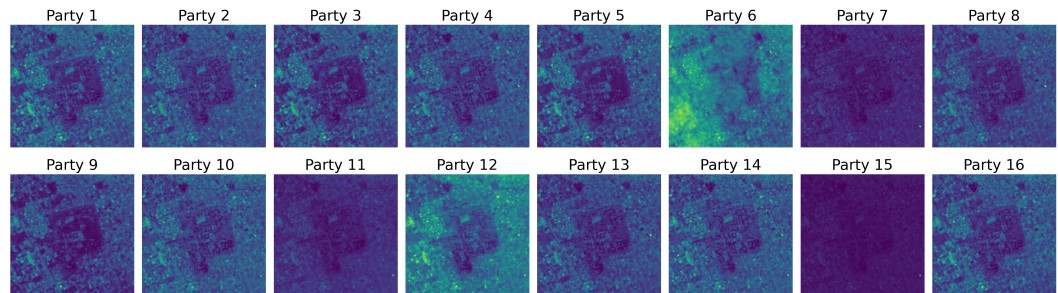

Figure 11: Preview of an area of `Satellite` dataset at channel-1 with land type Amnesty

Table 3: Channel information of satellite dataset

| Channel | Band | Description |
| --- | --- | --- |
| 1 | Band 1 | Aerosol (443 nm) |
| 2 | Band 2 | Blue (490 nm) |
| 3 | Band 3 | Green (560 nm) |
| 4 | Band 4 | Red (665 nm) |
| 5 | Band 5 | Vegetation Red Edge (705 nm) |
| 6 | Band 6 | Vegetation Red Edge (740 nm) |
| 7 | Band 7 | Vegetation Red Edge (783 nm) |
| 8 | Band 8 | NIR (842 nm) |
| 9 | Band 8A | Narrow NIR (865 nm) |
| 10 | Band 9 | Water Vapor (940 nm) |
| 11 | Band 10 | SWIR - Cirrus (1375 nm) |
| 12 | Band 11 | SWIR (1610 nm) |
| 13 | Band 12 | SWIR (2190 nm) |

**License.** Our use of the WorldStrat dataset was restricted to the labels and Sentinel-2 imagery, falling under the CC BY 4.0 (Commons, 2023a) license, while excluding high-resolution imagery that falls under the CC BY-NC 4.0 (Commons, 2023b) license. Therefore, we have released the `Satellite` dataset under the CC BY 4.0 license.

**File description and maintenance plan.** We aim to create a dedicated website for federated learning datasets to host `Satellite` and future VFL datasets. Although the website is currently under construction, we have made `Satellite` available via a public Google Drive link (Anonymized, 2023) for review purposes. The provided ZIP file comprises 32 CSV files, corresponding to training and testing datasets split at a ratio of 8:2. Each training and testing file contains 3,142 and 785

flattened images from a party, respectively. The code of VertiBench (Wu et al., 2023a) also contains a `Satellite` loader for demonstration.

## F  EXPERIMENTAL DETAILS

**Datasets.**  The datasets employed in our experiments exhibit a range of dimensions (from 16 to 20,958), instance numbers (from 15k to 581k), and tasks, which include binary classification, multiclass classification, and regression. Detailed information about these datasets and the corresponding licenses are presented in Table 4 and Table 5.

Table 4: Detailed information of centralized datasets (N/A means unspecified)

| Dataset | Task | #samples | #features | #classes | License |
|---|---|---|---|---|---|
| covtype (Blackard, 1998) | cls | 581,012 | 54 | 7 | BSD[1] |
| msd (Bertin-Mahieux, 2011) | reg | 463,715 | 90 | 1 | CC BY-NC-SA 2.0[2] |
| gisette (Guyon et al., 2008) | cls | 6,000 | 5,000 | 2 | CC BY 4.0[3] |
| realsim (Andrew, 2015) | cls | 72,309 | 20,958 | 2 | N/A |
| epsilon (Guo-Xun et al., 2008) | cls | 400,000 | 2,000 | 2 | N/A |
| letter (Slate, 1991) | cls | 15,000 | 16 | 26 | N/A |
| radar (Khosravi, 2020) | cls | 325,834 | 174 | 7 | N/A |
| MNIST (Deng, 2012) | cls | 70,000 | 784 | 10 | CC BY-SA 4.0 DEED[4] |
| CIFAR10 (Krizhevsky and Hinton, 2009) | cls | 60,000 | 1,024 | 10 | N/A |

[1] `https://opensource.org/license/bsd-3-clause/`
[2] `https://creativecommons.org/licenses/by-nc-sa/2.0/`
[3] `https://creativecommons.org/licenses/by/4.0/legalcode`
[4] `https://creativecommons.org/licenses/by-sa/4.0/deed.en`

Table 5: Detailed information of real VFL datasets

| Dataset | Task | #parties | #samples[1] | #features[2] | #classes |
|---|---|---|---|---|---|
| NUS-WIDE (Chua et al., 2009) | cls | 5 | 107,859/161,789 | 64/144/73/128/225 | 2 |
| Vehicle (Duarte and Hu, 2004) | cls | 2 | 63,058/15,765 | 50/50 | 3 |
| Satellite (proposed) | cls | 16 | $3,927\times16$ | $16\times13\times158\times158$ | 4 |

[1] The `#samples` is written in the form of `train/test`.
[2] The `#features` for each parties are divided by `/`.
The licenses of `NUS-WIDE` and `Vehicle` are unspecified. The license of `Satellite` is CC-BY-4.0

**Hyperparameters.**  For models based on split-GBDT, such as SecureBoost, FedTree, and Pivot, our experiments are conducted with the following hyperparameters: `learning_rate=0.1`, `num_trees=50`, `max_bin=32`, and `max_depth=6`. Due to the constraints of dataset sizes in their codes, Pivot is evaluated exclusively on two datasets: the `letter` dataset under the default setting of `MAX_GLOBAL_SPLIT_NUM=6,000` and on the `gisette` dataset with `MAX_GLOBAL_SPLIT_NUM=500,000`. The latter alteration was necessitated by a segmentation fault encountered under the default setting.

With regard to split-NN-based models, specifically SplitNN and C-VFL, each local model is trained by a two-layer multi-layer perceptron (MLP) with each hidden layer containing 100 units. The corresponding aggregated model is a single-layer MLP with 200 hidden units. The learning rate, chosen from the set $\{10^{-4}, 10^{-3}, 3 \times 10^{-3}\}$, is contingent on the specific algorithm and dataset. The number of iterations is fixed at 50 for SplitNN and 200 for C-VFL, with the latter setting aimed at ensuring model convergence. We also test C-VFL using four quantization buckets, a single vector quantization dimension, and a top-k compressor as recommended in the default setting. The number of local rounds $Q$ in C-VFL is set to 10. In the evaluation of communication cost, the split parameter $\alpha$ is set to 0.1 since feature split hardly affects the communication size.

Finally, for the ensemble-based model, GAL, we utilize a `learning_rate=0.01`, `local_epoch=20`, `global_epoch=20`, and `batch_size=512`, with the assist mode set

to `stack`. In the GAL framework, each party employs an MLP model consisting of two hidden layers, each containing 100 hidden units.

**Environments.** The hardware configuration used for C-VFL, GAL, SplitNN, and FedTree consists of 2x AMD EPYC 7543 32-Core Processors, 4x A100 GPUs, and 503.4 GB of RAM, running on Python 3.10.11 with PyTorch 2.0.0, Linux 5.15.0-71-generic, Ubuntu 22.04.2 LTS. For FATE framework, we are using `federatedai/standalone_fate` Docker image, running with Python 3.8.13 on Docker 23.0.2. Pivot is compiled from source using CMake 3.19.7, g++ 9.5.0, libboost 1.71.0, libscapi with git commit hash `1f70a88`, and runs on a slurm cluster with AMD EPYC 7V13 64-Core Processor with the same number of cores as 2x AMD EPYC 7543 used for other algorithms. We conducted real distributed experiments on four distinct machines to evaluate the communication cost, the machines are equipped with AMD EPYC 7V13 64-Core Processors, 503Gi RAM, 1GbE NICs (Intel I350), and varying GPUs (AMD Instinct MI210 for $P_1$ and $P_2$, AMD Instinct MI100 for $P_3$ and $P_4$). The distributed experimental environment consisted of Python 3.9.12 and PyTorch 2.0.1+rocm5.4.2. We monitored network traffic from the network adapter before and after the experiments using `ifconfig` to measure the actual data size transferred between parties. We also recorded the network traffic size during idle server periods as background noise. Measurements were taken for 1 minute with a 1-minute gap between each measurement, totaling 5 times. The mean received data was 56.76 KB (std ±1.08 KB), and the mean sent data was 46.50 KB (std ±1.35 KB).

**Distributed Implementations.** FedTree inherently supports distributed deployment through gRPC, a feature we utilize. In contrast, SplitNN, C-VFL, and GAL lack native distributed support. To address this, we implemented distributed support for these algorithms using `torch.distributed`. For the sparse matrix representation in C-VFL, we employ `torch.sparse_coo_tensor`.

**License.** The licenses pertinent to the datasets and algorithms utilized in VertiBench are documented in Table 4 and Table 6, respectively. We ensure adherence to these licenses as VertiBench neither redistributes the codes and data nor utilizes them for any commercial purpose.

Table 6: License usage for each VFL algorithm.

| Algorithm | License | Share | Commercial use | Redistribute |
|---|---|---|---|---|
| GAL (Diao et al., 2022) | MIT[1] | ✓ | ✓ | ✓ |
| C-VFL (Castiglia et al., 2022) | MIT[1] | ✓ | ✓ | ✓ |
| SecureBoost (Cheng et al., 2021) | Apache 2.0[2] | ✓ | ✓ | ✓ |
| Pivot (Wu et al., 2020) | CC BY-NC-ND 4.0[3] | ✓ | ✗ | ✗ |
| FedTree (Li et al., 2023) | Apache 2.0[2] | ✓ | ✓ | ✓ |

Note: AL, SplitNN, BlindFL and VF2Boost do not specify their licensing terms.
[1] https://opensource.org/license/mit/
[2] https://www.apache.org/licenses/LICENSE-2.0
[3] https://creativecommons.org/licenses/by-nc-nd/4.0/

# G ADDITIONAL EXPERIMENTS

## G.1 COMMUNICATION COST DETAILS

In this subsection, we evaluate the total and maximum incoming/outgoing communication costs of various VFL algorithms over 50 epochs. This assessment is performed on four physically distributed machines, with results displayed in Figures 12, and 13. We observe real network adapter communication for both four and two-party settings with `ifconfig`. Additionally, theoretically estimated communication sizes are compared in Figure 14, and real-time performance in relation to increasing communication costs for four VFL algorithms is presented in Figure 15. The results yield following four observations.

**Gradient-boosting algorithms, such as GAL and FedTree, tend to have small communication sizes when contrasted with NN-based methods like SplitNN and C-VFL. Yet, FedTree experiences marked communication overhead with high-dimensional datasets.** C-VFL, due to its

different pipeline from SplitNN, fails to reduce real communication cost, especially as it broadcasts compressed representations from all parties too all other parties. The large communication expenses in NN-based algorithms from the recurrent transmission of gradients and representations, contributing to SplitNN's superior accuracy. The high communication costs of FedTree on high-dimensional datasets might be linked to the transmission of feature histograms and node information across parties.

**Generally, GAL is more efficient than FedTree, though FedTree sometimes has less incoming communication cost on low-dimensional datasets.** The incoming communication cost includes residuals for GAL and histograms for FedTree. Residual size is dependent on the number of instances, while histogram size relates to the feature count. Hence, FedTree shows small incoming communication cost on low-dimensional datasets but more on high-dimensional datasets. However, FedTree's outgoing communication cost is much larger than GAL, as it transmits both gradients and node information, resulting in its large overall communication overhead.

**C-VFL struggles to reduce real communication costs via representation compression sent to the server, and it neither enhance the backward server-client communication.** Initially, C-VFL's incoming communication costs align closely with those of SplitNN. This is attributable to the method of estimating the compression ratio of representations by counting the ratio of non-zero elements. Though these representations appear sparse, they cannot be effectively transmitted without added overheads. Specifically, the coordinate format (COO) sparse matrix, as utilized in our experiments, demands even greater communication due to the need for additional indices. As a result, such compressions offer limited advantage in practical scenarios. Moreover, the consistent outgoing communication costs indicate that transmitting compressed aggregated models in C-VFL or broadcasting compressed representations does not display efficiency over transmitting uncompressed gradients of the cut layer. This revelation hints at avenues for potential refinement to minimize SplitNN's backward communication expenses.

**The real-time performance per communication round aligns with observations made for a fixed epoch.** This is largely attributed to the dominance of communication per iteration over the number of iterations. GAL achieve similar accuracy levels with minimal communication overhead, a scenario where SplitNN and C-VFL might struggle to complete even a single iteration. However, when more communication is allowed, SplitNN typically delivers the most superior performance.

**The estimated communication costs generally align with actual observations, with notable exceptions in C-VFL and FedTree when handling high-dimensional datasets.** In the case of C-VFL, the estimation overlooks the overhead associated with transmitting sparse matrices. For FedTree on high-dimensional data, the reported communication cost likely arises from unaccounted transmission information related to nodes and histograms. Such discrepancies underscore the need for comprehensive testing on real distributed systems for VFL algorithms.

## G.2 SCALABILITY

In this section, we examine the scalability of various VFL algorithms on two high-dimensional datasets, depicted in Figure 16. The datasets, split by importance with $\alpha = 1$, consist of a varying number of parties, ranging from 2 to 2048. Our results demonstrate that **SplitNN and FedTree are scalable to thousands of parties without any significant drop in accuracy.** This is attributable to FedTree's lossless design and SplitNN's robust structure. However, both GAL and C-VFL show substantial performance declines with an increase in party numbers.

An intriguing observation is that **C-VFL's accuracy nearly matches SplitNN's when the number of parties reaches 2048 on the `gisette` dataset**. This is likely because the average number of features per party reduces to 2 in this scenario, causing the compression mechanism to potentially fail, and thus, C-VFL reverts to SplitNN's performance.

## G.3 TRAINING TIME

The training duration for VFL algorithms is consolidated in Table 7. It should be noted that FedTree and SecureBoost are executed without the use of encryption or noise. Conversely, we retain the default privacy setting for Pivot as it does not offer a non-encryption alternative. Three observations can be gleaned from the table.

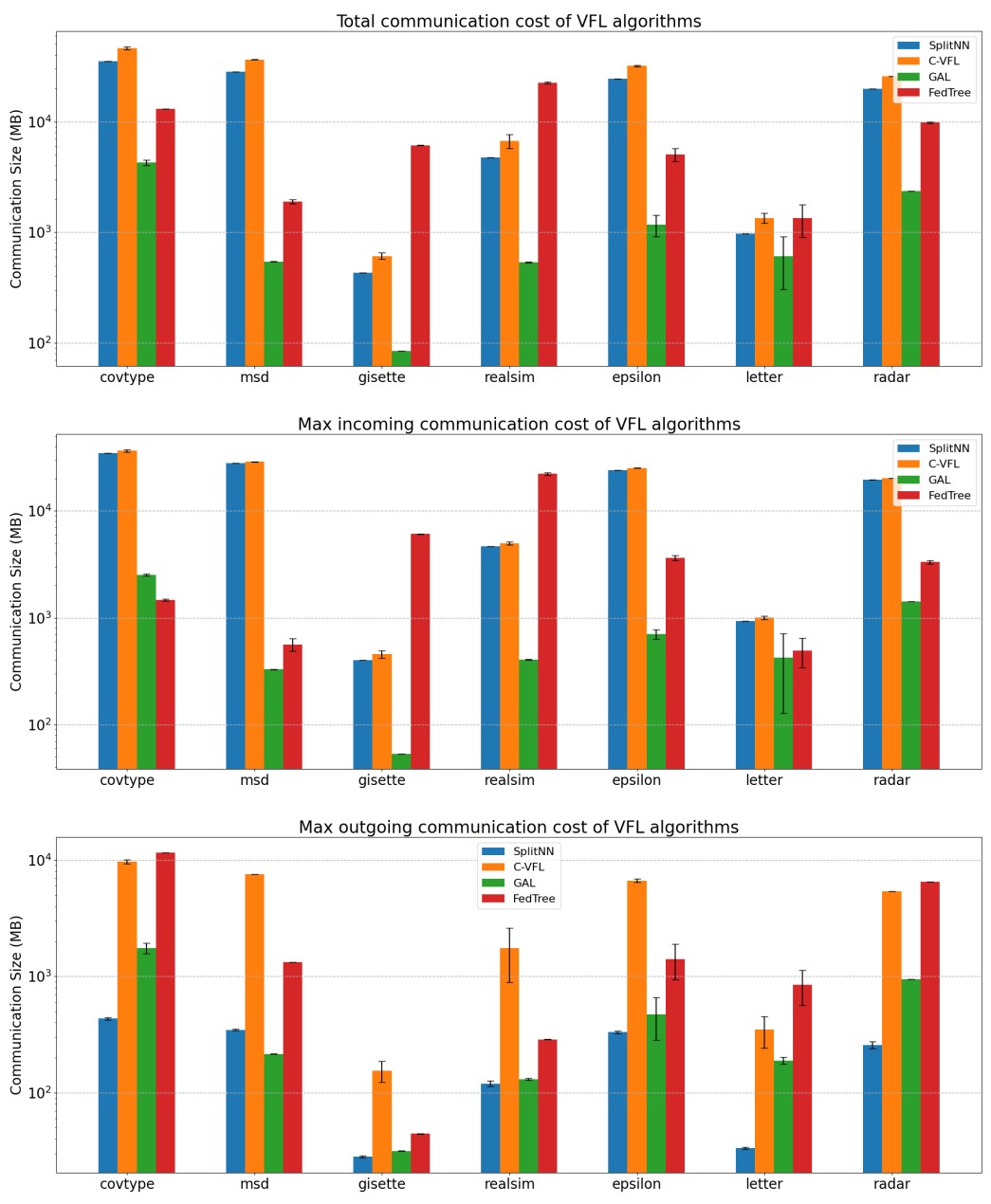

Figure 12: Real total, max incoming and outgoing communication size across parties of VFL algorithms (50 global iterations)

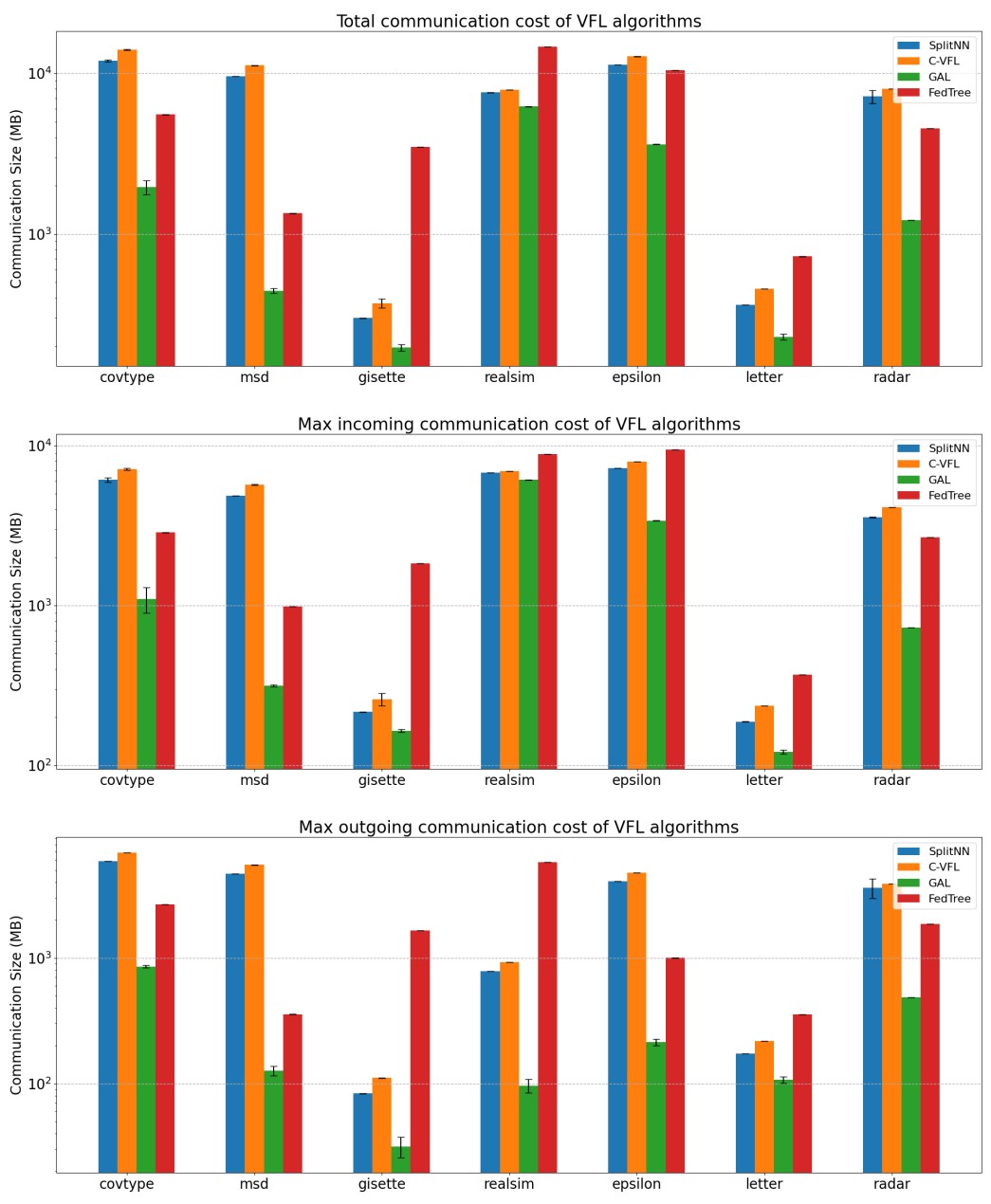

Figure 13: Real total, max incoming and outgoing communication size across parties of VFL algorithms (two parties, 50 global iterations)

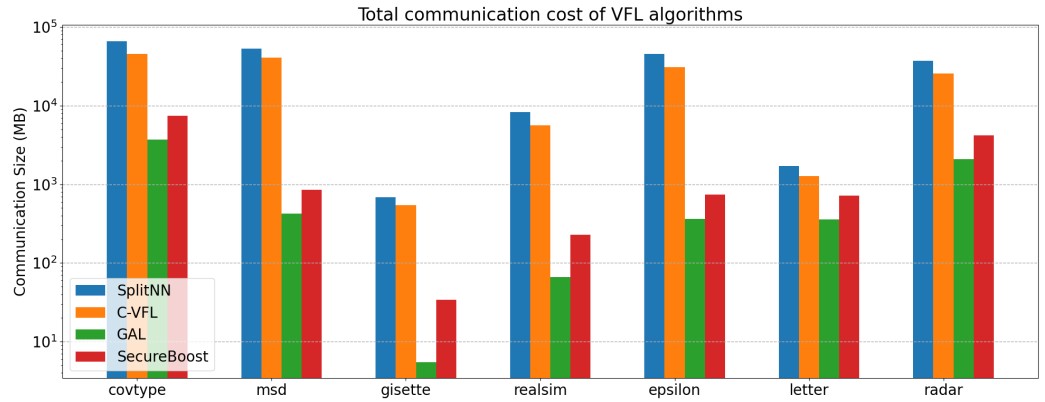

Figure 14: Estimated total communication size of VFL algorithms (50 global iterations)

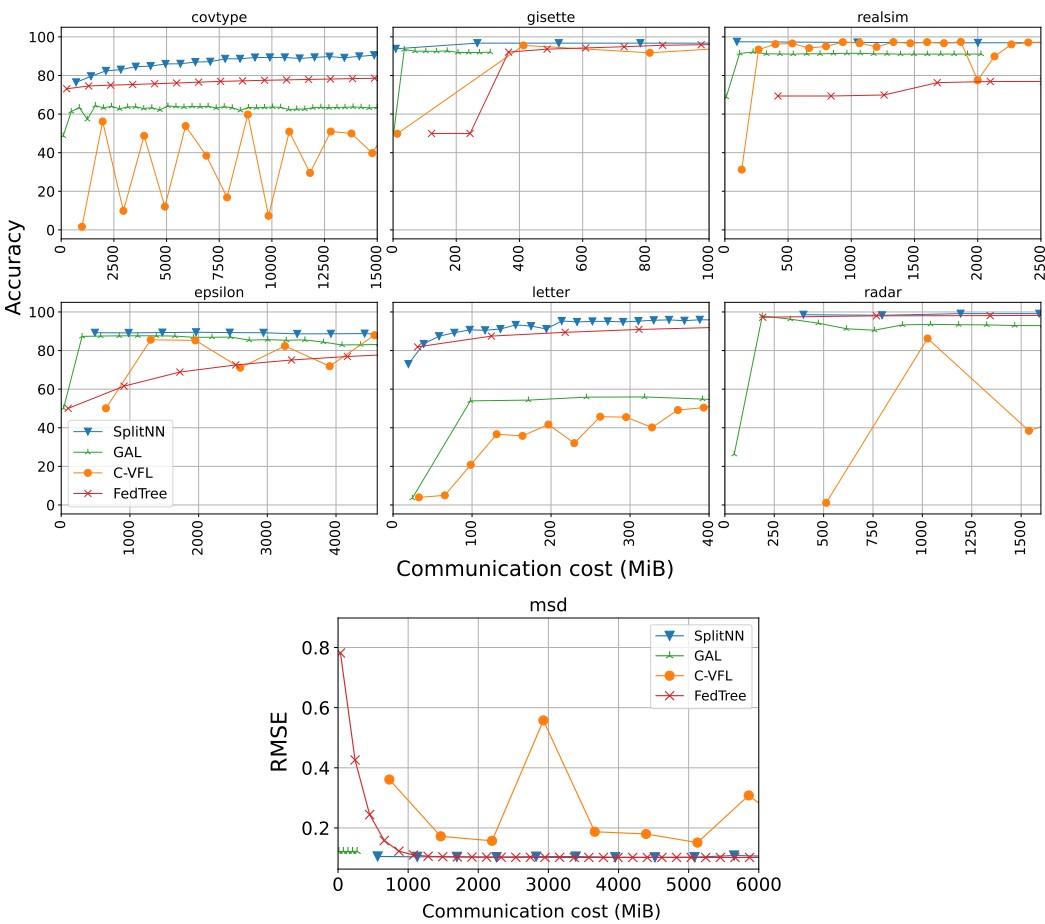

Figure 15: Test performance as the increasing of communication cost (MiB)

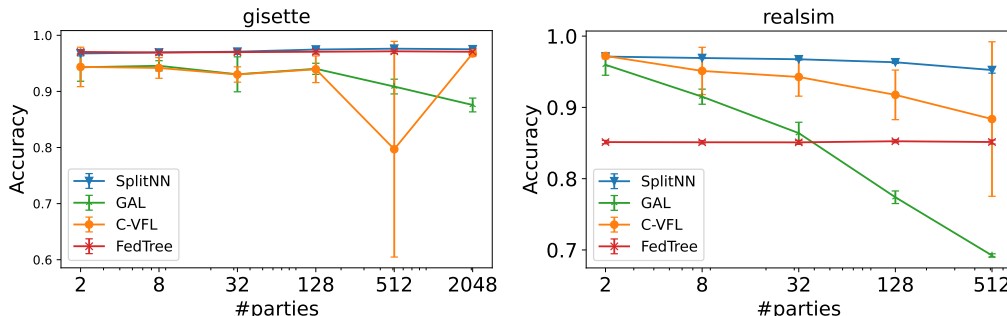

Figure 16: Scalability tests for VFL algorithms on `gisette` and `realsim` dataset

Firstly, **we observe a considerable overhead associated with the encryption processes of Pivot**. Pivot, which employs both homomorphic encryption and secure multi-party computation to ensure stringent privacy, endures a training time that is up to $10^5$ times longer than FedTree. This limitation renders such strict privacy measures impractical for real-world applications that employ large datasets. This observation underscores the necessity for further exploration into the efficiency-privacy trade-off in VFL.

Secondly, when comparing non-encryption methods, we find that **split-based algorithms (SplitNN, FedTree) generally outperform ensemble-based algorithms (GAL) in terms of efficiency**. This is primarily because split-based algorithms require each party to train a partial model, whereas ensemble-based algorithms mandate that each party train an entire model for ensemble purposes. This design characteristic also contributes to the lower communication costs associated with ensemble-based algorithms, as demonstrated in Figure 12.

Lastly, we note that **SplitNN demonstrates higher efficiency than FedTree on high-dimensional small datasets, yet demands more training time on low-dimensional large datasets**. This discrepancy arises because FedTree computes a fixed-size histogram for each feature, which alleviates the impact of a large number of instances but is sensitive to the number of features. Conversely, SplitNN trains data in batches, rendering it sensitive to the number of instances. This observation emphasizes the importance of carefully selecting a VFL algorithm based on the properties of the dataset in the application.

Table 7: Estimated training time of VFL algorithms (4-party, in hours)

| Dataset | VFL Algorithm | | | | | |
|---------|---------|------|---------|-------------|-------|----------|
|         | SplitNN | GAL | FedTree | SecureBoost | C-VFL | Pivot [1] |
| covtype | 0.19 | 2.18 | 0.076 | 14.84 | 1.45 | - |
| msd     | 0.17 | 2.01 | 0.037 | 0.28  | 1.29 | - |
| gisette | 0.01 | 0.03 | 0.052 | 0.56  | 0.02 | 25805.52 |
| realsim | 0.06 | 1.40 | 0.32  | 1.16  | 0.48 | - |
| epsilon | 0.15 | 1.79 | 0.33  | 7.99  | 1.11 | - |
| letter  | 0.01 | 0.06 | 0.47  | 2.69  | 0.04 | 6750.76 |
| radar   | 0.12 | 1.31 | 0.21  | 10.06 | 0.83 | - |

[1] A segmentation fault arises in Pivot when datasets encompass more than 60,000 samples. Consequently, only the `gisette` and `letter` datasets have their training times estimated.

### G.4    PERFORMANCE ON SATELLITE DATASET

In Table 8, we present the single-party and VFL performance results on the Satellite dataset. For an equitable comparison, each single party trains a concatenated MLP, formed by linking a SplitNN's local model with its aggregated model, under the same hyperparameters. Our results indicate that

VFL can yield approximately a 10% accuracy improvement over local training, thus affirming the practical utility of the Satellite dataset for vertical federated learning applications.

Table 8: Accuracy of single party training (Solo) and VFL with SplitNN on Satellite dataset

| Method | Accuracy (mean $\pm$ standard deviation) |
|---|---|
| Party $P_0$ (Solo) | $0.7368 \pm 0.0086$ |
| Party $P_1$ (Solo) | $0.7310 \pm 0.0054$ |
| Party $P_2$ (Solo) | $0.7320 \pm 0.0055$ |
| Party $P_3$ (Solo) | $0.7289 \pm 0.0115$ |
| Party $P_4$ (Solo) | $0.7251 \pm 0.0080$ |
| Party $P_5$ (Solo) | $0.7167 \pm 0.0097$ |
| Party $P_6$ (Solo) | $0.7353 \pm 0.0045$ |
| Party $P_7$ (Solo) | $0.7256 \pm 0.0139$ |
| Party $P_8$ (Solo) | $0.7335 \pm 0.0126$ |
| Party $P_9$ (Solo) | $0.7154 \pm 0.0049$ |
| Party $P_{10}$ (Solo) | $0.7264 \pm 0.0172$ |
| Party $P_{11}$ (Solo) | $0.7169 \pm 0.0106$ |
| Party $P_{12}$ (Solo) | $0.7182 \pm 0.0075$ |
| Party $P_{13}$ (Solo) | $0.7118 \pm 0.0084$ |
| Party $P_{14}$ (Solo) | $0.7060 \pm 0.0119$ |
| Party $P_{15}$ (Solo) | $0.7292 \pm 0.0069$ |
| Party $P_1 \sim P_{15}$ (SplitNN) | $\mathbf{0.8117 \pm 0.0035}$ |

### G.5 Details of VFL Performance

In this subsection, we present the detailed information (including msd) in Table 9. Besides the previous observations, we can make two additional observations on the `msd` regression dataset. First, compared among algorithms, split-based algorithms still have leading performance. Second, compared among split parameters, we observe that both $\alpha$ and $\beta$ have slight effect on the performance on `msd`.

### G.6 Ablation Study of Truncated Threshold of SVD

In this subsection, we assess the Pcor of each dataset using VertiBench, varying the truncated threshold $d_t$. We quantify both the speedup - defined as the execution time ratio between exact and truncated SVD - and the relative error, as outlined in Eq. 37.

$$\text{Relative Error} = \frac{|\text{Approximate Pcor} - \text{Exact Pcor}|}{\text{Exact Pcor}} \quad (37)$$

From the presented data, it is evident that high-dimensional datasets such as `gisette`, `realsim`, `epsilon`, and `radar` benefit from notable speedup with minimal error. In contrast, for lower-dimensional datasets like `covtype`, `msd`, and `letter`, the speedup is marginal and accompanied by a significant relative error. Consequently, in our experiments, we employ exact SVD for datasets comprising fewer than 100 features. For those exceeding 100 features, we utilize approximate SVD with $d_t = 400$.

### G.7 Performance on MNIST and CIFAR10

In this subsection, we extend VertiBench to accommodate image datasets, with experiments conducted on `MNIST` and `CIFAR10` revealing novel insights.

**Image Dataset Splitting.**   Unlike tabular datasets, image datasets hold crucial positional information, implying that feature order is pertinent. Adapting to this characteristic, we flatten the image and perform the split using VertiBench, akin to tabular datasets, all while preserving positional metadata. Post-splitting, the features are reconfigured to their original image positions, and any absent features

Table 9: Accuracy/RMSE of VFL algorithms on different datasets varying imbalance and correlation

| Dataset | Method | Performance of importance-based split | | | | Performance of correlation-based split | | | |
|---|---|---|---|---|---|---|---|---|---|
| | | $\alpha = 0.1$ | $\alpha = 1$ | $\alpha = 10$ | $\alpha = 100$ | $\beta = 0$ | $\beta = 0.3$ | $\beta = 0.6$ | $\beta = 1$ |
| covtype | SplitNN | **91.2±0.4%** | **92.1±0.2%** | **92.1±0.3%** | **92.1±0.1%** | **92.0±0.2%** | **92.1±0.2%** | **92.3±0.2%** | **92.1±0.1%** |
| | GAL | 66.1±2.9% | 60.2±1.1% | 62.1±4.1% | 61.5±3.9% | 63.1±2.3% | 62.9±2.4% | 63.0±2.4% | 64.0±1.6% |
| | FedTree | 77.9±0.1% | 77.8±0.2% | 77.8±0.1% | 77.8±0.1% | 77.8±0.2% | 77.8±0.2% | 77.8±0.2% | 77.9±0.2% |
| | C-VFL | 46.9±4.1% | 14.8±19.0% | 26.9±25.3% | 38.3±18.6% | 26.6±18.2% | 36.3±17.4% | 39.2±15.4% | 46.6±6.7% |
| | FedOnce | 68.9±5.9% | 70.2±2.6% | 74.4±3.1% | 75.0±1.5% | 71.6±3.5% | 73.4±1.7% | 75.9±0.8% | 73.8±2.5% |
| msd | SplitNN | **0.1010±0.0** | **0.1020±0.0** | **0.1010±0.0** | **0.1010±0.0** | **0.1015±0.0** | **0.1015±0.0** | **0.1013±0.0** | **0.1010±0.0** |
| | GAL | 0.1220±0.0 | 0.1220±0.0 | 0.1220±0.0 | 0.1220±0.0 | 0.1222±0.0 | 0.1222±0.0 | 0.1222±0.0 | 0.1222±0.0 |
| | FedTree | 0.1040±0.0 | 0.1040±0.0 | 0.1040±0.0 | 0.1040±0.0 | 0.1035±0.0 | 0.1035±0.0 | 0.1035±0.0 | 0.1035±0.0 |
| | C-VFL | 0.1270±0.0 | 0.1270±0.0 | 0.1270±0.0 | 0.1270±0.0 | 0.1910±0.0 | 0.1708±0.0 | 0.1878±0.0 | 0.1830±0.0 |
| | FedOnce | 0.1110±0.0 | 0.1103±0.0 | 0.1080±0.0 | 0.1079±0.0 | 0.1100±0.0 | 0.1097±0.0 | 0.1089±0.0 | 0.1092±0.0 |
| gisette | SplitNN | 96.8±0.4% | 96.8±0.3% | 96.9±0.3% | 96.9±0.2% | **97.0±0.4%** | 96.9±0.6% | 96.9±0.3% | 96.7±0.3% |
| | GAL | 94.4±1.7% | 95.7±0.6% | 96.2±0.6% | 96.1±0.6% | 95.4±0.5% | 95.5±0.5% | 95.6±0.5% | 95.4±0.6% |
| | FedTree | **97.0±0.3%** | 97.1±0.5% | **97.1±0.5%** | **97.1±0.3%** | **97.0±0.4%** | **96.9±0.5%** | **97.0±0.3%** | **97.1±0.4%** |
| | C-VFL | 91.0±11.5% | **97.8±3.4%** | 90.1±11.8% | 94.0±4.0% | 95.0±1.9% | 95.7±1.2% | 95.0±1.2% | 94.6±2.0% |
| | FedOnce | 91.0±9.8% | 96.4±0.6% | 96.3±0.5% | 96.2±0.5% | 95.6±0.4% | 95.6±0.5% | 96.0±0.7% | 95.7±0.5% |
| realsim | SplitNN | **97.1±0.0%** | **97.0±0.0%** | **97.0±0.1%** | **97.1±0.1%** | **96.9±0.2%** | **97.0±0.2%** | **97.0±0.1%** | **96.9±0.2%** |
| | GAL | 92.4±0.7% | 93.5±2.2% | 96.1±0.4% | 96.4±0.1% | 96.5±0.2% | 96.5±0.2% | 96.5±0.2% | 96.5±0.2% |
| | FedTree | 85.1±0.2% | 85.2±0.2% | 85.1±0.1% | 85.1±0.2% | 85.2±0.2% | 85.2±0.3% | 85.2±0.2% | 85.1±0.2% |
| | C-VFL | 89.6±17.4% | 88.6±16.7% | 89.5±17.4% | 87.8±16.1% | 95.9±2.1% | 96.5±0.8% | 94.9±3.6% | 93.8±3.7% |
| | FedOnce | 94.1±3.0% | 95.1±0.4% | 95.6±0.3% | 95.2±0.2% | 94.9±0.3% | 94.9±0.3% | 95.1±0.3% | 94.7±0.2% |
| epsilon | SplitNN | 86.3±0.1% | **86.2±0.1%** | 86.3±0.0% | 86.2±0.0% | 85.9±0.1% | 86.0±0.1% | 86.1±0.1% | 86.2±0.1% |
| | GAL | **88.2±0.6%** | 85.9±1.5% | 85.7±0.4% | **86.3±0.2%** | 85.8±0.4% | 85.8±0.4% | 85.8±0.4% | 85.8±0.4% |
| | FedTree | 77.2±0.1% | 77.3±0.1% | 77.2±0.0% | 77.2±0.1% | 77.2±0.1% | 77.2±0.1% | 77.3±0.0% | 77.2±0.1% |
| | C-VFL | 50.1±0.0% | 53.5±4.9% | 67.7±20.0% | 61.6±16.4% | 73.8±7.5% | 65.0±11.1% | 71.3±13.5% | 69.3±14.8% |
| | FedOnce | 76.6±12.8% | 79.5±3.0% | 81.7±3.0% | 80.7±1.7% | 79.7±1.2% | 80.1±1.6% | 78.3±2.0% | 77.7±1.0% |
| letter | SplitNN | **95.5±0.3%** | **96.0±0.3%** | **96.1±0.2%** | **96.0±0.3%** | **95.0±0.3%** | **94.7±0.4%** | **94.7±0.4%** | **95.0±0.3%** |
| | GAL | 57.9±4.1% | 51.4±3.1% | 52.0±4.3% | 49.1±2.6% | 48.9±1.9% | 48.9±1.9% | 48.9±1.8% | 49.0±1.9% |
| | FedTree | 91.7±0.3% | 91.9±0.4% | 92.0±0.3% | 92.0±0.3% | 91.8±0.3% | 91.9±0.4% | 91.9±0.3% | 92.0±0.3% |
| | C-VFL | 5.0±1.0% | 12.6±8.3% | 37.1±47.1% | 43.8±4.5% | 69.4±4.2% | 70.4±2.4% | 70.6±3.7% | 68.1±5.7% |
| | FedOnce | 50.3±9.6% | 55.6±3.9% | 60.1±1.9% | 60.0±1.1% | 60.2±2.1% | 59.8±3.6% | 59.1±2.1% | 60.4±2.2% |
| radar | SplitNN | **99.7±0.0%** | **99.8±0.0%** | **99.8±0.0%** | **99.8±0.0%** | **99.8±0.0%** | **99.8±0.0%** | **99.8±0.0%** | **99.8±0.0%** |
| | GAL | 90.0±3.6% | 95.9±2.5% | 97.5±0.4% | 97.9±0.1% | 96.5±0.1% | 96.5±0.1% | 96.5±0.1% | 96.5±0.1% |
| | FedTree | 99.3±0.0% | 99.3±0.0% | 99.3±0.0% | 99.3±0.0% | 99.3±0.0% | 99.3±0.0% | 99.3±0.0% | 99.2±0.0% |
| | C-VFL | 59.9±47.8% | 53.4±38.5% | 43.1±23.9% | 37.3±15.7% | 35.5±18.6% | 44.5±17.9% | 53.2±19.8% | 40.0±15.0% |
| | FedOnce | 97.7±1.6% | 98.4±0.2% | 98.4±0.2% | 98.4±0.1% | 98.1±0.1% | 98.1±0.1% | 98.1±0.1% | 98.3±0.1% |

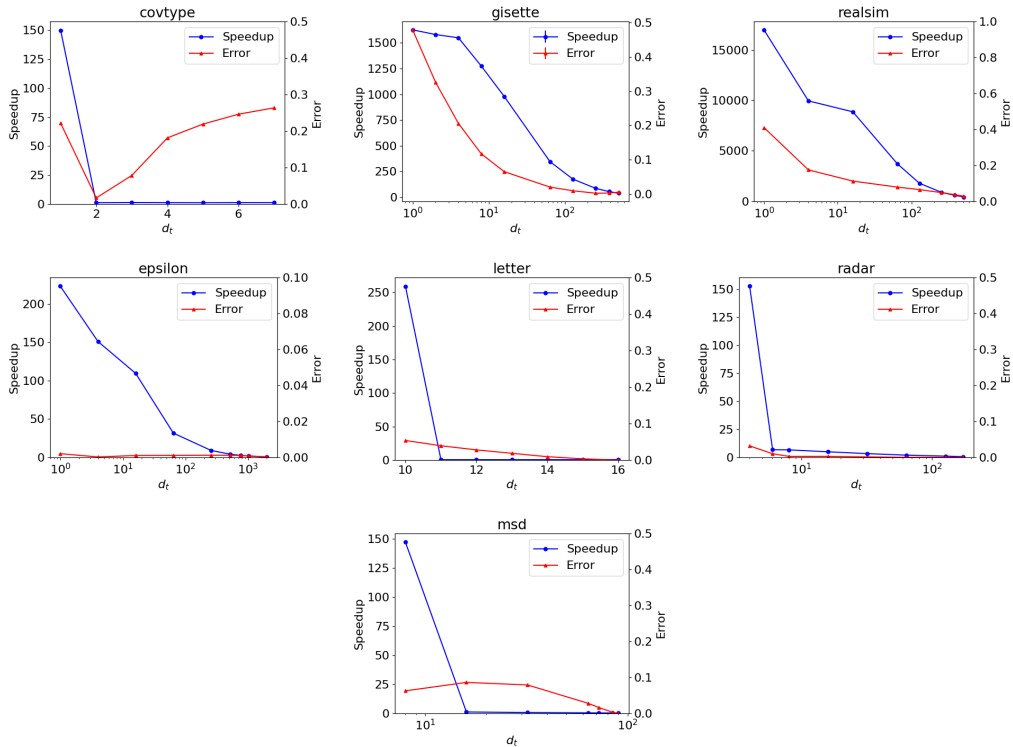

Figure 17: Effect of $d_t$ on relative error and speedup

are replaced with the background color - black for `MNIST` (Figure 20,21) and white for `CIFAR10` (Figure 18,19).

**Visualization Insights.** The processed images post-split elucidate the effects of $\alpha$ and $\beta$. Analyzing by party importance, $\alpha$, we observe that smaller $\alpha$ values result in certain parties retaining recognizable image details (e.g., party 2), while others contain little meaningful information. Conversely, a larger $\alpha$ distributes image data more equitably across parties. Addressing party correlation, $\beta$, especially in the `MNIST` context, it is observed that a smaller $\beta$ ensures each party is allocated distinct parts of an image. For instance, with $\beta = 0.0$, party 3 predominantly possesses the left portion of the digit "8", while party 4 obtains its right portion. In contrast, a larger $\beta$ induces high inter-party correlation, with all parties vaguely representing the digit. This trend is less pronounced in `CIFAR10` due to its narrower Icor range.

**Experiments.** In our experimentation on image datasets, we utilize the ResNet18 architecture. Figure 22 presents the outcomes and suggests three new observations:

- **GAL consistently has leading performance, particularly on `CIFAR10`.** The performance of GAL can be attributed to the relatively large number of parameters of ResNet18. In contrast, the use of split-NN-based algorithms, which concatenate models, may introduce excessive parameters leading to overfitting.

- **On imbalanced VFL datasets, the efficacy of GAL is reduced.** This decrease in performance might be due to less-informative parties introducing noise during the boosting process.

- **C-VFL demonstrates a close performance to SplitNN in image classification tasks**, which is potentially rooted in the sparsity of image representations.

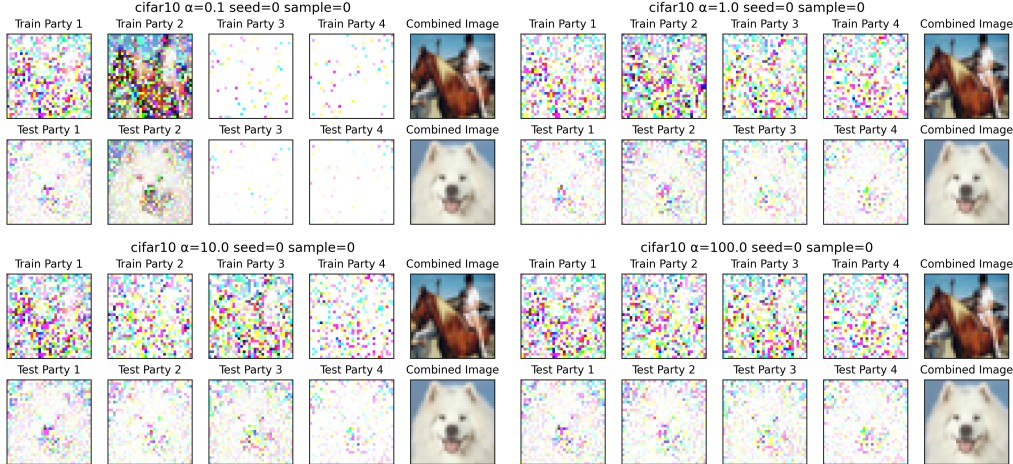

Figure 18: Visualization of CIFAR10 split by different $\alpha$

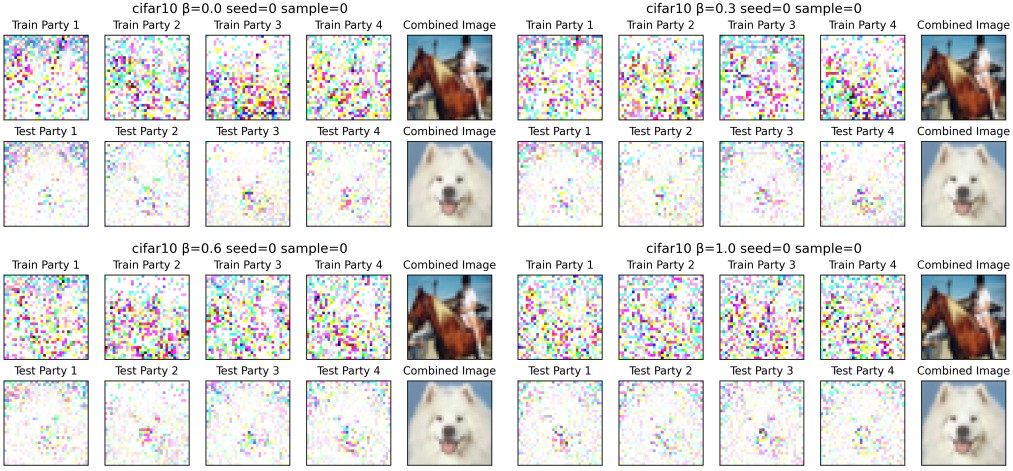

Figure 19: Visualization of CIFAR10 split by different $\beta$

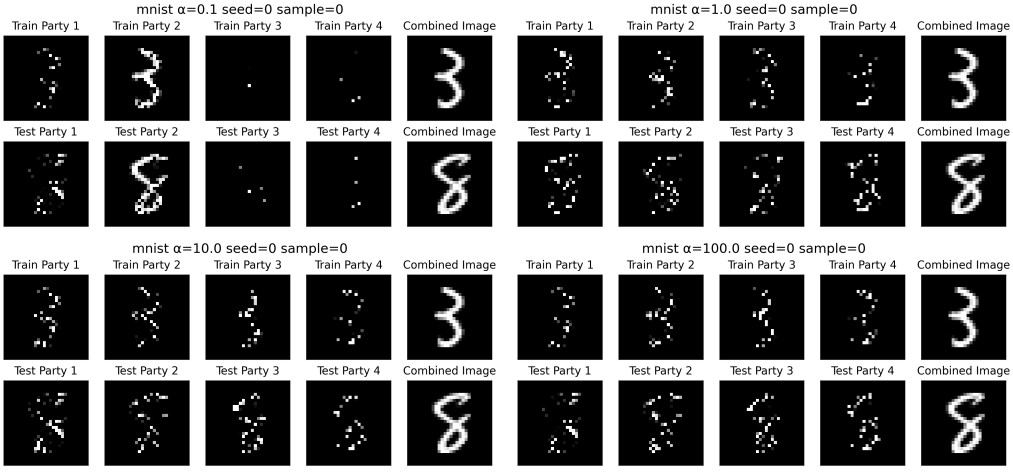

Figure 20: Visualization of MNIST split by different $\alpha$

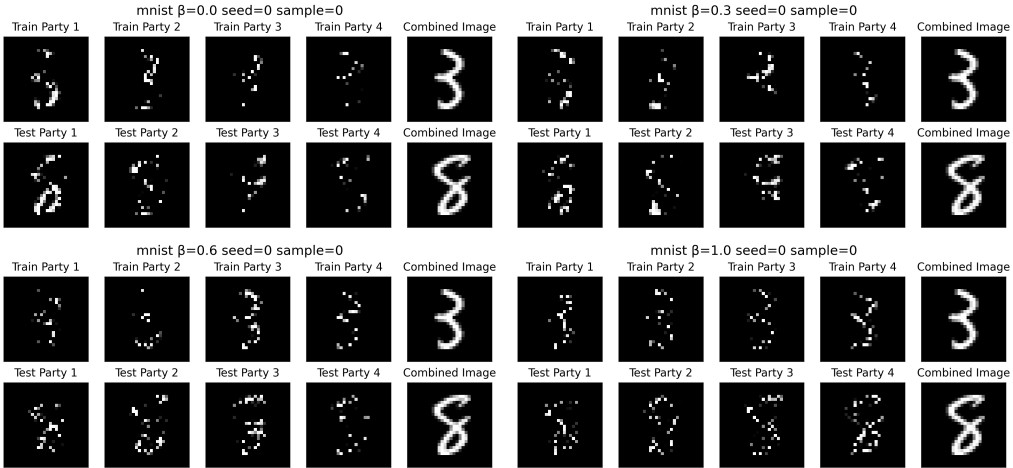

Figure 21: Visualization of MNIST split by different $\beta$

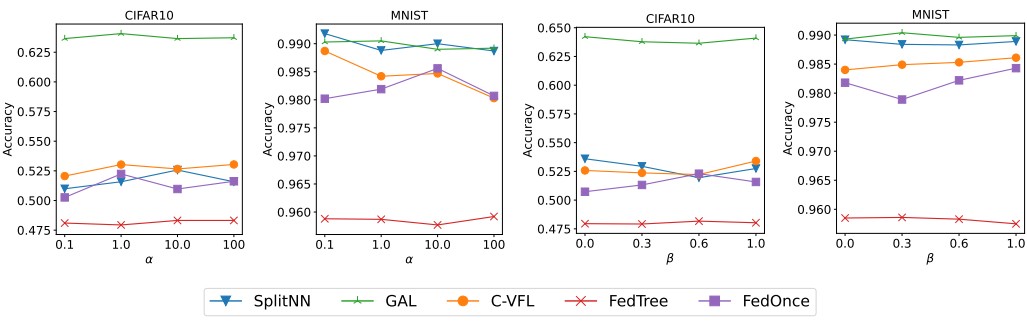

Figure 22: The accuracy of MNIST and CIFAR10 using ResNet18 varying imbalance($\alpha$) and correlation($\beta$)

G.8  PERFORMANCE CORRELATION: SYNTHETIC VS. REAL VFL DATASETS (ADDITIONAL)

In this subsection, we extend the experiments of Section 4.4 and showcase the detailed performance on each dataset on real and synthetic datasets in Figure 23. Our experiments yield two primary observations, suggesting the alignment of VeritBench scope and the real scope. Firstly, the relative performance ranking of VFL algorithms remains consistent across real and synthetic datasets. Secondly, for algorithms sensitive to data splits like VFL, the trends observed on synthetic datasets generally align with real ones. For instance, C-VFL exhibits superior performance on the balanced two-party `Vehicle` dataset compared to the imbalanced five-party `NUS-WIDE`. Similarly, it fares better on balanced two-party synthetic data than on imbalanced five-party datasets, a pattern consistent across synthetic datasets derived from both `covtype` and `letter`.

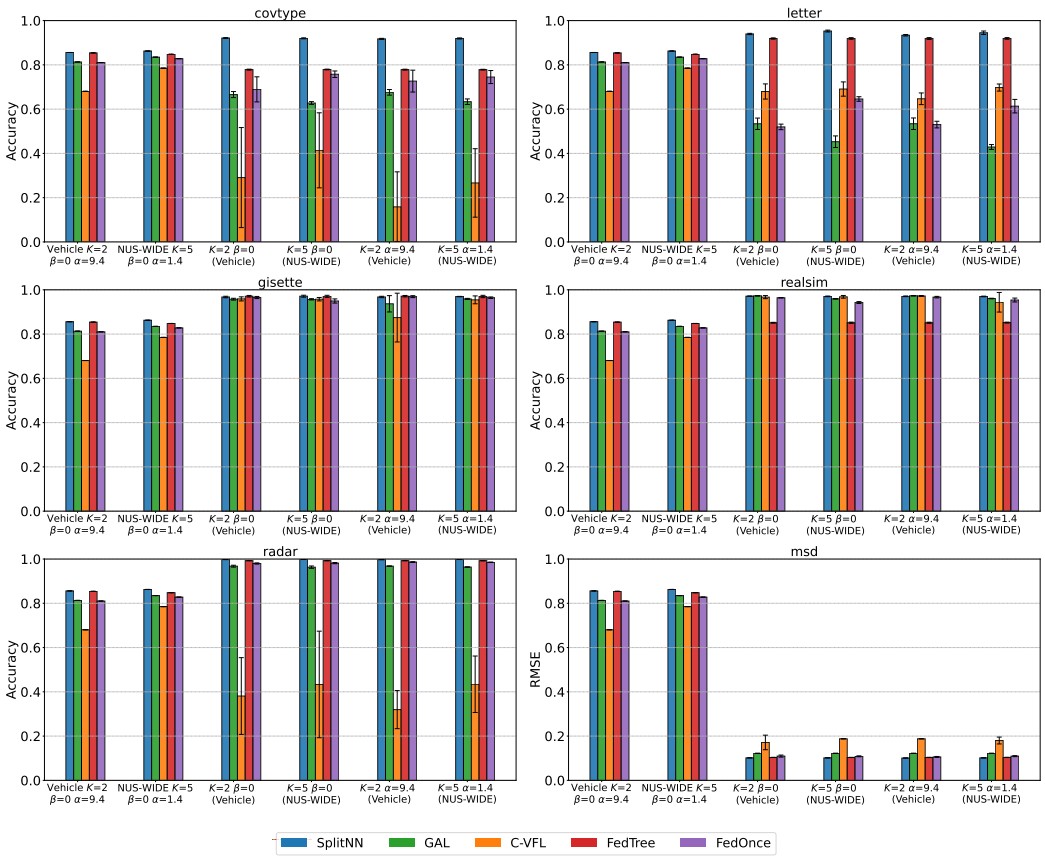

Figure 23: Performance of VFL algorithms on real vs. synthetic datasets with consistent $\alpha, \beta$

G.9  VARIANCE OF PERFORMANCE

In our experiment, we examined the performance variance of VFL algorithms under a consistent $\alpha$ or $\beta$, revealing that VFL accuracy is affected by the randomness of feature partitioning. Figures 24 investigates the convergence of both mean and variance in algorithm performance as the number of test repetitions increased. Figure 25 illustrate the performance distributions under the same $\alpha$ or $\beta$.

Figure 24 reveals that both mean and variance stabilize after 5-10 iterations, showcasing distinct performance variance across different algorithms. Notably, C-VFL displays the highest variance, while SplitNN and FedTree demonstrate consistently low variance, underscoring the algorithm-dependent nature of performance variance. Additionally, Figure 25 illustrates that most algorithms exhibit a concentrated distribution of results.

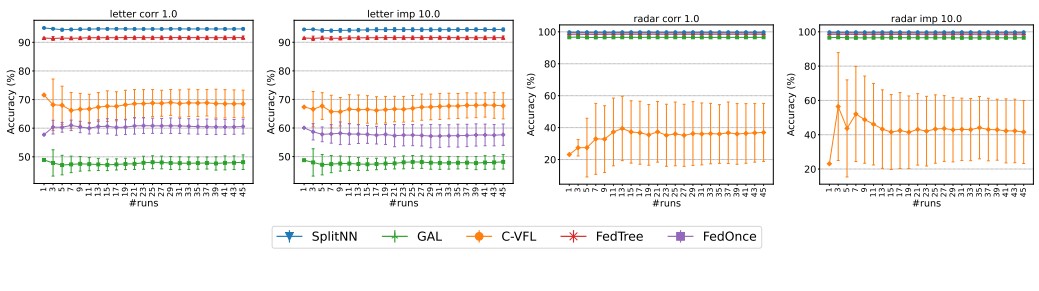

Figure 24: Standard deviation of accuracy as number of seeds tested

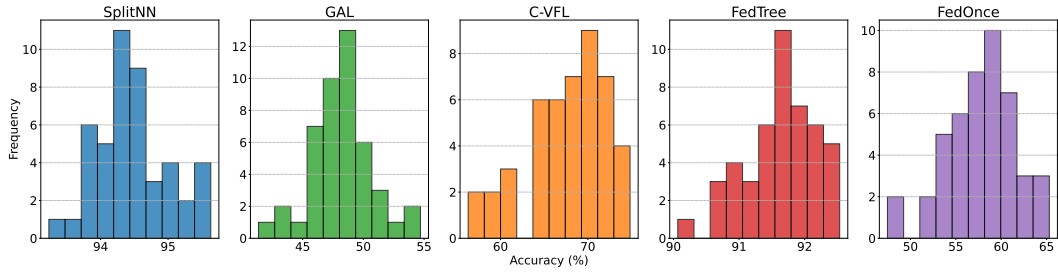

(a) letter when $\alpha = 10.0$

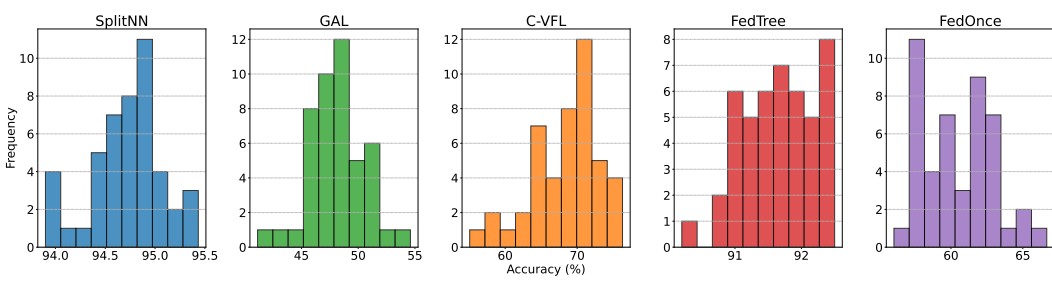

(b) letter when $\beta = 1.0$

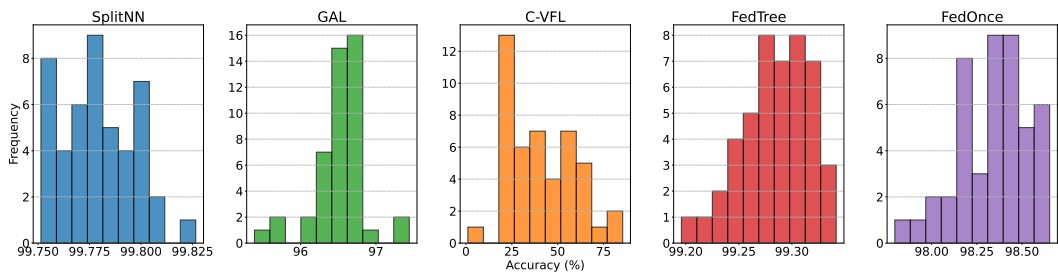

(c) radar when $\alpha = 10.0$

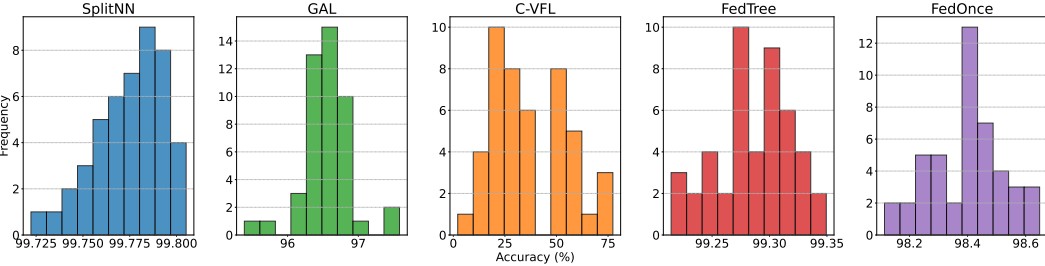

(d) radar when $\beta = 1.0$

Figure 25: Accuracy distribution of VFL algorithms

In comparing Spearman rank and Pearson correlation coefficients, we find Spearman rank effective for non-linear correlations, while Pearson is suitable for linear correlations. Figure 26 illustrates the performance differences on datasets split using these metrics.

For linearly correlated datasets like `gisette`, Pearson's coefficient is preferable. It accurately captures the enhanced performance of C-VFL on highly correlated splits, a nuance Spearman rank-based splitting misses. Conversely, for non-linearly correlated datasets such as `radar`, Spearman rank-based correlation is more effective, highlighting variations in C-VFL performance that Pearson-based splits overlook. Therefore, the choice of correlation coefficient should align with the dataset's correlation characteristics.

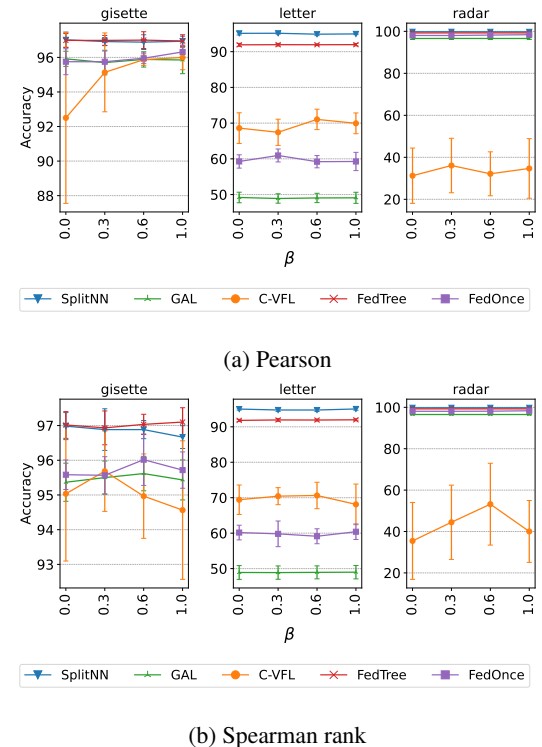

(a) Pearson

(b) Spearman rank

Figure 26: Accuracy of VFL algorithms on different datasets varying correlation metrics

# H   DISCUSSION

This section discusses the limitations of VertiBench, shedding light on several areas where improvement is needed. Additionally, we engage in a discussion surrounding potential negative social impacts and personal privacy issues related to VertiBench.

## H.1   LIMITATIONS

In this subsection, we outline the limitations of VertiBench, focusing on three primary aspects.

**Scalability of correlation-based split.**   The correlation-based split method that we propose may face efficacy and efficiency challenges when applied to a large number of parties. As the number of parties increases, the potential feature splits proliferate exponentially. This complexity presents a significant obstacle for optimization methods such as BRKGA (Gonçalves and Resende, 2011), making it challenging to locate the minimum and maximum Icor, as well as the optimal split that corresponds to the given $\beta$. This situation underscores the necessity for more advanced permutation-

based optimization algorithms that can enable the correlation-based split method to scale out to a greater number of parties.

**Relationship between importance and correlation.** Within VertiBench, we regard importance and correlation as two orthogonal factors impacting the feature split. However, this viewpoint might overlook the potential correlation that could exist between these two factors. For instance, in cases of highly imbalanced feature split, parties might demonstrate low inter-party correlation. As a result, a comprehensive benchmarking framework that simultaneously considers both importance and correlation is desired to provide a more rigorous evaluation of VFL algorithms.

Designing a benchmark that captures the intricate relationship between feature importance and correlation presents significant challenges, as certain levels of correlation and importance may be mutually exclusive. Designing a co-optimization algorithm for both feature importance and correlation while maintaining efficiency and explainability presents a significant challenge. Balancing the demands of these metrics could lead to compromises in both computational efficiency and the explainability of the resulting data splits. Developing an approach that effectively integrates these factors without sacrificing these key aspects remains a complex, yet crucial task in advancing VFL benchmarking methodologies.

**Evaluation of privacy.** Although VertiBench assesses performance, efficiency, and communication cost, it does not provide a quantitative evaluation of privacy. The high performance observed with SplitNN could potentially come at the cost of privacy, while the markedly high overhead of Pivot might be attributed to its robust privacy requirements. The task of quantitatively evaluating the privacy of different VFL algorithms and models remains an open problem, which we aim to tackle in future work.

## H.2 SOCIAL IMPACTS

**Negative social impact.** While VertiBench primarily focuses on analyzing and comparing existing methodologies, and hence is less likely to cause additional negative social impact, the potential for biased interpretation of our experimental results could inadvertently mislead future research or applications. Specifically, we emphasize that the superior performance of non-encrypted methods such as SplitNN and GAL does not necessarily indicate that they are fit for immediate deployment in real-world VFL applications. The privacy concerns arising from the transfer of residuals or representations require further investigations. A quantitative benchmark on privacy is a critical prerequisite to deploying VFL approaches in real-world applications, which we plan to explore in the future research.

