# OpenReview forum: "VertiBench: Advancing Feature Distribution Diversity in Vertical Federated Learning Benchmarks"
_ICLR.cc/2024/Conference — ICLR 2024 poster_

### Official Review · Reviewer_6Ygd · 2023-10-21

**Soundness:** 2 fair
**Presentation:** 3 good
**Contribution:** 3 good
**Rating:** 6
**Confidence:** 2

**Summary:**

This research paper proposes improvements to vertical federated learning (VFL) benchmarks by addressing the shortcomings of existing benchmarks. The paper introduces feature importance and correlation as two crucial factors that could potentially influence the performance of VFL algorithms. It also proposes evaluation metrics and dataset splitting methods to improve VFL performance. The study evaluates cutting-edge VFL algorithms, including ensemble-based, Split-NN-based, and Split-GBDT-based algorithms, and provides insights gained from the evaluation. Overall, the paper provides valuable insights for future research in the field of VFL.

**Strengths:**

1.	This paper is well-written, so I can pick the core ideas up effortlessly.
2.	The focused problem has broad interests, as VFL has attracted much attention from both academia and industry, but there are rarely practical or realistic datasets. This paper improves existing VFL benchmarks by addressing the shortcomings of inadequate evaluation of existing VFL methods.
3.	This paper introduces feature importance and correlation as crucial factors that could influence VFL algorithm performance. Meanwhile, it proposes evaluation metrics and dataset splitting methods to improve VFL performance.
4.	Additionally, the evaluation of cutting-edge VFL algorithms provides valuable insights for future research in the field.

**Weaknesses:**

1.	The notation needs to be clarified. In Equation 1, the summation is over index $i$, but the added terms do not contain $i$. This forbids me to understand the meaning of this decomposition of the left-hand side’s log probability.
2.	From another point of view, as synthetic datasets can be split in different ways, focusing on either importance or correlations, real-world datasets are still in more demanded. Thus, offering a more realistic VFL dataset in a related benchmark would be better.
3.	The proposed correlation-based split method may face scalability challenges when applied to a large number of parties, which could limit its practicality in certain scenarios. Could you provide more discussion about this?

**Questions:**

Is it possible to use one metric for reflecting both importance and correlation? Or say how should a user trade-off between these two factors in comparing different VFL algorithms?

---

> ### Author Response · Authors · 2023-11-21
> **Response to Reviewer 6Ygd**
>
> We are grateful for the insightful feedback from the reviewers and have thoroughly revised our manuscript in line with your suggestions. Below, we detail how each concern and question has been addressed. We welcome any further feedback and kindly request your reconsideration of our revised submission, including a reassessment of its score if deemed appropriate.
>
> 1. _"The notation needs to be clarified."_
>
> We have fixed this issue in Equation 1. This index should be $k$ instead of $i$.
>
> 2. _"Offering a more realistic VFL dataset in a related benchmark would be better"_
>
> In response to the need for a broader range of real-world VFL datasets, we have developed a new image-to-image VFL dataset, named "Satellite." This dataset fills a crucial gap, as there currently are no image-image VFL datasets available in the existing benchmarks like NUS-WIDE, Vehicle, and FedAds. The construction process and specifics of the Satellite dataset are comprehensively detailed in Appendix E. Additionally, we have conducted evaluations using SplitNN on this dataset, presented in Appendix G.4, to demonstrate its potential for improvement in VFL scenarios. This initiative not only enhances the diversity of our VFL benchmarks but also contributes a novel dataset to the field, broadening the scope for future research and application.
>
> >In this section, we outline the construction process of the Satellite dataset, which was adapted from the WorldStrat dataset~\cite{satellite}, originally intended for high-resolution imagery analysis. The Satellite dataset encompasses Point of Interest (POI) data, each associated with one or more Areas of Interest (AOI). Every AOI incorporates a unique location identifier, a land type, and 16 low-resolution, 13-channel images, each taken during a satellite visit to the location.
>
> In our revised manuscript, we acknowledge the limitations of expanding the real scope due to dataset size constraints. To mitigate this, we provide additional insights in Section 4.4, demonstrating that VertiBench-synthetic datasets effectively represent the relative performance of VFL algorithms on real-world datasets. This approach not only broadens the applicability of our benchmarks but also offers a practical solution for evaluating VFL algorithms in scenarios where large-scale real datasets are not readily available.
>
> 3. _"correlation-based split method may face scalability challenges when applied to a large number of parties, which could limit its practicality in certain scenarios."_
>
> In Appendix D.3, we provide detailed timings for feature splits across datasets, encompassing a range of up to 0.58 million instances and 20,000 features. Additionally, we conducted scalability tests for the correlation-based split method on synthetic data of varying sizes.
>
> The importance-based method proves to be highly efficient, completing within 30 seconds across all datasets tested. This method, which randomly splits without necessitating the computation of Shapley values, still retains some Shapley-related properties, ensuring scalability for large datasets.
>
> Conversely, the correlation-based method, while not as rapid, is sufficiently efficient for practical applications. Our scalability tests demonstrate that this method can handle up to 10,000 features, with the time cost increasing approximately linearly with the number of features. This level of efficiency is adequate for most datasets typically used in VFL, such as realsim and CIFAR10.
>
> We emphasize that the permutation-based optimization in VertiBench is an orthogonal problem. Should more advanced optimization algorithms become available, they can be seamlessly integrated to enhance VertiBench's efficiency.

---

> ### Author Response · Authors · 2023-11-22
> **Response for Reviewer 6Ygd (Additional Response to Questions)**
>
> 4. _"Is it possible to use one metric for reflecting both importance and correlation? Or say how should a user trade-off between these two factors in comparing different VFL algorithms?"_
>
> In our revised Section 2.1 and Appendix H.1, we clarify the rationale behind choosing specific split methods depending on the scenario.
>
> >For datasets with nearly independent features, the low inter-party correlation makes correlation-based splits less meaningful, suggesting the superiority of importance-based feature splits. Conversely, in datasets with highly correlated features, assessing individual feature importance becomes impractical, making correlation-based splits more suitable due to varying inter-party correlations.
> >
> >Importance and correlation are treated as orthogonal evaluation factors applicable in distinct scenarios. While there may be an intrinsic link between them, our experiments indicate that focusing on one factor at a time yields explainable results reflective of real-world performance. As discussed in Appendix H.1, the interplay between importance and correlation can be complex. A joint optimization for both factors might be computationally intensive and less explainable, while providing limited additional insights.
>
> Essentially, splitting based on individual factors offers two key advantages:
>
> - **Efficiency**: Joint optimization of multiple factors is substantially more complex and resource-intensive compared to optimizing a single factor.
>
> - **Explainability**: Separate factor optimization allows for clearer insights into VFL algorithm performance. For instance, as the balance parameter $\alpha$ increases, GAL's performance tends to drop while FedOnce improves. This suggests GAL is better suited for imbalanced datasets and FedOnce for balanced ones. This inference aligns with real-world VFL dataset performances. In the relatively imbalanced Vehicle dataset (small $\alpha$), GAL outperforms FedOnce by 2%. Conversely, in the NUS-WIDE dataset, GAL's performance is comparable to FedOnce, differing by less than 0.3%.
>
> Furthermore, although VertiBench doesn't support simultaneous factor-based splitting, it is capable of evaluating both importance and correlation effectively. This is exemplified in Figure 2, showcasing its capability in assessing both factors concurrently.

---

> ### Author Response · Authors · 2023-11-22
> **Clarification on the Response to Concern 3**
>
> We acknowledge the misunderstanding of the concern (3) regarding the scalability of the correlation split method in relation to party size. To address this, we conducted additional experiments illustrated in Figure 10c (Appendix D.3), focusing on the impact of increasing party numbers on efficiency. The results show that within a fixed global dataset, the number of parties has minimal effect on the efficiency of the correlation split method. We trust this experiment more accurately addresses your concerns.

---

### Official Review · Reviewer_BWae · 2023-10-31

**Soundness:** 3 good
**Presentation:** 2 fair
**Contribution:** 3 good
**Rating:** 8
**Confidence:** 4

**Summary:**

This paper tackles the problem of inadequate feature diversity in VFL benchmarking. Motivated from a probabilistic perspective, this paper factors the feature diversity into party importance and party correlation.

Based on this factorization, this paper proposes corresponding metrics for real VFL datasets (*Shapley-Based metric for party importance, SVD-based Icor metric for party correlation*) and a tailored splitting method for synthesized VFL datasets (*Dirichlet-based importance splitting and permutation-based correlation splitting*). By further mapping the two evaluation metrics into splitting parameters $\alpha$ and $\beta$, the paper provides a unified solution to enable cross-scope comparison between real and synthesized datasets.

Empirical experiments are conducted on 2 real VFL datasets and 7 centralized datasets. The results validate the effectiveness of the designed metrics and split method, and provide additional observations to better understand 4 representative VFL methods.

**Strengths:**

- **Tackling an important problem**: Feature diversity is a fundamental property that highly affects the performance of VFL. Evaluating VFL algorithms across different levels of feature distribution contributes to understanding the distribution feasibility of a proposed method. In HFL, using the Dirichlet distribution to simulate various Non-IID levels has already become a consensus and common tool in HFL algorithm evaluation. It's happy to see this paper provides an effective tool for VFL to achieve a similar purpose. This would be helpful in forming a more fair and comprehensive standard in benchmarking VFL methods.
- **Effectiveness of the Method is Mostly Supported**: The basic purposes of Party Importance and Party Correlation are well-achieved as shown in Figures 2, 3, and 4. The splitting method shows ideal results as expected. Besides, the importance splitting method has a good theoretical guarantee. It shows promise to become a commonly utilized tool in future VFL research, thereby making a significant impact.
- **Sufficient Benchmark of Major VFL Method Type**: This is the first work comparing three different classes of VFL methods (AL-based, NN-based, and Tree-based), providing insights from a broader view. The experiments are conducted on sufficient datasets with various settings. The appendix provides very detailed additional results, and the splitting method used in the CIFAR10 and MNIST datasets is much more reasonable than that in previous works (left-right image half split).

**Weaknesses:**

- **Unclear Independent Assumption**: From the beginning (the end of Section 2.1), the reason for omitting the correlation between Party Importance and Party Correlation is not well-described. A rough or intuitive relationship would be better to explain such a choice (e.g., the independent assumption is more concise and sufficient for evaluating most cases, or explain why it is so hard to capture their correlation). Since there are two metrics representing feature diversity, it can be confusing to determine which one to use, which one is better suited in specific cases, and whether it is possible and reasonable to evaluate a specific method in a 2D grid of $\alpha$ and $\beta$.
- **Seemingly Case-Restricted Significance**: As shown in Figure 5, only C-VFL reflects a significant response to the variation of feature diversity, and all other methods perform consistently robust on all datasets. It seems like there are some specific reasons tailored to C-VFL that cause this observation, and further, **it makes me doubt whether it is meaningful to conduct such feature diversity evaluation for non-compression-based VFL methods**. This may largely reduce the potential impact of this work. Besides, as shown in the right part of Figure 7, the performance rank of GAL observed on synthetic datasets is different from real ones and shows significantly low performance. The consistency level is not as strong as claimed by words.

**Questions:**

I am willing to increase my score if the authors adequately address the questions or clarify my misunderstandings.

**Major**

See in the Weakness part.

**Minor**
- **Unclarities**:
  - It is confusing when reading "VertiBench, when parameterized with the same $\beta$, exactly reconstruct the real feature split of Vehicle dataset." in Section 3.2 and comparing Figure 4 (a) vs (b), since you do not introduce how we can get $\alpha$ and $\beta$ for real VFL datasets. There should be a kind hint to remind readers to refer to Section 3.3 for details.
  - There are 2 real VFL datasets and 7 centralized datasets used in the experiment. However, the introduction of the two real VFL datasets is missing, it should be clearly described in section 4.2. This confused me a lot at the beginning.
  - The citation for Shapley-CMI is likely incorrect in line 5 of Section 2.2; the given citation is about the "Shapley Taylor Interaction Index". However, the correct one is given in Line 3 after Theorem 1.
- **Citation Absence or Overclaim**: As is commonly expected of benchmark papers, they are usually anticipated to provide a comprehensive literature review of related methods in this domain and clarify their connection with existing benchmark papers. However, the discussion of a recent benchmark paper, [1]FedAds, is omitted, and some splitNN-based methods (e.g., [2]FedHSSL, [3]JPL, and others listed in Table 3 of work[1]) as well as other types of tree-based methods (e.g., [4]Federated Forest) are omitted in discussion. Although it is not necessary to evaluate all these methods, a comprehensive literature review is usually beneficial for a benchmark paper to be solid. From this perspective, it would be more appropriate to restrict it as a benchmark **only for major VFL method types**, since the paper excludes some sub-type methods.
  - [1] FedAds: A Benchmark for Privacy-Preserving CVR Estimation with Vertical Federated Learning
  - [2] A Hybrid Self-Supervised Learning Framework for Vertical Federated Learning
  - [3] Vertical Semi-Federated Learning for Efficient Online Advertising
  - [4] Federated Forest

---

> ### Author Response · Authors · 2023-11-21
> **Response to Reviewer BWae (Part 1)**
>
> We are grateful for the insightful feedback from the reviewers and have thoroughly revised our manuscript in line with your suggestions. Below, we detail how each concern and question has been addressed. We welcome any further feedback and kindly request your reconsideration of our revised submission, including a reassessment of its score if deemed appropriate.
>
> 1. _"Unclear Independent Assumption"_
> In our revised Section 2.1 and Appendix H.1, we clarify the rationale behind choosing specific split methods depending on the scenario.
>
> >For datasets with nearly independent features, the low inter-party correlation makes correlation-based splits less meaningful, suggesting the superiority of importance-based feature splits. Conversely, in datasets with highly correlated features, assessing individual feature importance becomes impractical, making correlation-based splits more suitable due to varying inter-party correlations.
> >
> >Importance and correlation are treated as orthogonal evaluation factors applicable in distinct scenarios. While there may be an intrinsic link between them, our experiments indicate that focusing on one factor at a time yields explainable results reflective of real-world performance. As discussed in Appendix H.1, the interplay between importance and correlation can be complex. A joint optimization for both factors might be computationally intensive and less explainable, while providing limited additional insights.
>
> Essentially, splitting based on individual factors offers two key advantages:
>
> - **Efficiency**: Joint optimization of multiple factors is substantially more complex and resource-intensive compared to optimizing a single factor.
>
> - **Explainability**: Separate factor optimization allows for clearer insights into VFL algorithm performance. For instance, as the balance parameter $\alpha$ increases, GAL's performance tends to drop while FedOnce improves. This suggests GAL is better suited for imbalanced datasets and FedOnce for balanced ones. This inference aligns with real-world VFL dataset performances. In the relatively imbalanced Vehicle dataset (small $\alpha$), GAL outperforms FedOnce by 2%. Conversely, in the NUS-WIDE dataset, GAL's performance is comparable to FedOnce, differing by less than 0.3%.
>
> Furthermore, although VertiBench doesn't support simultaneous factor-based splitting, it is capable of evaluating both importance and correlation effectively. This is exemplified in Figure 2, showcasing its capability in assessing both factors concurrently.
>
> 2. _"**Seemingly Case-Restricted Significance**"_
>
> To address concerns about the effect of $\alpha,\beta$, we introduced a new baseline, FedOnce, significantly influenced by split factors. For the detailed values of the performance shifts of other VFL algorithms, which may be overshadowed by C-VFL's variability, refer to Table 9. Here's a summary of how each algorithm is affected by $\alpha$ and $\beta$:
>
> - C-VFL: Performance varies up to 40% with $\alpha$ and up to 20% with $\beta$.
> - FedOnce: Shows a 10% variation with $\alpha$ on the letter dataset, and 5-7% on covtype, gisette, and epsilon; affected by 1-3% with $\beta$ on covtype and epsilon.
> - GAL: Impacted by 8% with $\alpha$ on letter and radar, and 2-5% on *all* other datasets; minimal impact from $\beta$.
> - SplitNN: Shows minimal variation due to robustness (though at a higher communication cost).
> - FedTree/SecureBoost: Not affected by feature split, theoretically matching centralized training performance.
>
> Notably, on the letter dataset, the performance ranking between FedOnce and GAL switches as $\alpha$ increases from 0.1 to 100. This suggests that GAL is better suited for imbalanced VFL scenarios, while FedOnce excels in balanced splits.
>
> This trend aligns with real VFL datasets: For the relatively imbalanced Vehicle dataset (small $\alpha$), GAL outperforms FedOnce by 2%. In contrast, on the NUS-WIDE dataset, both GAL and FedOnce show closely matched performance (within 0.3% difference), reinforcing that these synthetic dataset conclusions can effectively mirror real-world scenarios.

---

> ### Author Response · Authors · 2023-11-21
> **Response to Reviewer BWae (Part 2)**
>
> 3. _"It is confusing when reading "VertiBench ..."_
>
> We have moved this part to Appendix D.2 according to Reviewer myAZ's advice.
>
> 4. _"The introduction of the two real VFL datasets is missing"_
>
> We acknowledge the earlier confusion and have now added clarifying introductions in both Section 4.2 and Appendix F for enhanced understanding.
>
> 5. _"Wrong citation of Shapley-CMI"_
>
> We have corrected this wrong citation.
>
> 6. _"Citation Absence or Overclaim"_
>
> In Appendix B, we have thoroughly discussed and cited all mentioned studies, including a new addition, the FedOnce algorithm, to our benchmark for deeper insights. We acknowledge the challenge of evaluating all existing VFL codes, primarily due to the unavailability of open-source versions of many algorithms. Our primary objective is to demonstrate the efficacy of VertiBench across various VFL types, rather than evaluate as many algorithms as possible. Recognizing the diverse subtypes within VFL, as indicated, we have stated this limitation in our introduction and Section 4.1.
>
> > In this paper, we concentrate on the primary types of VFL, acknowledging that there are various subtypes as identified in (Liu et al., 2022). Exploring these subtypes in depth will be an objective of our future research efforts.

---

> > ### Comment · Reviewer_BWae · 2023-11-23
> >
> > Thanks to the authors' response. I am satisfied to the answers to both major and minor concerns, which include a detailed explanation of correlation-importance disentanglement, new experimental results showing significance to various methods, and detailed changes to tackle minor issues.
> >
> > Practical and efficient evaluation criteria is benefit to the progress of VFL research. As the first comprehensive main-type VFL benchmark paper aimed at more robust utility evaluation, i would believe it'll promote future works.
> > I welcome continuous progress in this seemingly dry but important direction.
> >
> > For these reasons, I have raised my score to 'Accept'.

---

### Official Review · Reviewer_oXEA · 2023-11-01

**Soundness:** 2 fair
**Presentation:** 3 good
**Contribution:** 2 fair
**Rating:** 6
**Confidence:** 2

**Summary:**

The paper introduces "VERTIBENCH," a novel contribution aimed at advancing the state of benchmarking in the domain of Vertical Federated Learning (VFL). VFL is a crucial paradigm for machine learning on feature-partitioned, distributed data. However, the lack of public real-world VFL datasets, and the limited diversity of feature distributions in existing benchmarks, hinders algorithm evaluation. To address these limitations, the paper introduces two essential factors in VFL performance evaluation: feature importance and feature correlation. It proposes associated evaluation metrics and dataset splitting methods to enrich benchmark datasets and provide a more comprehensive assessment of VFL algorithms. The paper presents a thorough evaluation of state-of-the-art VFL algorithms, offering valuable insights for future research in this field.

**Strengths:**

1. The paper effectively addresses a critical limitation in the field of VFL benchmarking. The lack of real-world datasets and the limited diversity of feature distributions in existing benchmarks have been significant challenges. By introducing novel factors and associated evaluation metrics, the paper enriches benchmark datasets and provides a more robust framework for evaluating VFL algorithms.


2. The paper conducts a comprehensive evaluation of state-of-the-art VFL algorithms. This empirical assessment provides valuable insights into the performance of these algorithms and helps researchers and practitioners in making informed decisions regarding algorithm selection and development.

3. The proposed benchmarking framework and insights provided by the paper contribute to the advancement of research in the field of VFL. By addressing the need for more diverse and real-world benchmark datasets, the paper lays the foundation for future investigations and developments in this area.

**Weaknesses:**

1. The paper highlights, in Figure 1(b), the relatively small scope of existing VFL datasets. This raises questions about the alignment of the proposed VFL dataset synthesis approach with real-world scenarios. Rather than expanding the scope to what may be nearly impossible in practice, it might be more appropriate to delve deeper into the real-world scope. While I recognize that Section 4.5 includes experiments to demonstrate the relationship between synthetic and real-world VFL datasets, further clarification is needed on how to interpret the findings from Figure 7 to better understand the implications of the results.

2. Section 2.3 discusses various party correlation metrics, with the paper employing Spearman rank correlation in Algorithm 1. It would be beneficial to explore whether Spearman rank correlation can be replaced by the other mentioned correlations and, if possible, provide a comparison of their performance. This comparison would enhance the understanding of the selected correlation metric's effectiveness.

3. Figure 5 indicates that three out of four VFL baselines are minimally impacted by alpha and beta. This raises questions about the rationale for partitioning the dataset based on feature importance and party correlation. It prompts further consideration of whether the proposed VFL dataset synthesis approach may overemphasize expanding the scope to an extent that may not reflect practical scenarios. Clarifying the reasoning behind these observations is crucial.

4. The paper discusses the communication efficiency of VFL algorithms, which seems to be minimally impacted by data distribution. Given this, it might be more appropriate to allocate more space to extending the experiments in Section 4.5. Moving additional explanations and results, which may be included in the appendix, to the main paper, would provide a more comprehensive understanding of the proposed method and its real-world applicability.

**Questions:**

Please see weaknesses.

---

> ### Author Response · Authors · 2023-11-21
> **Response to Reviewer oXEA**
>
> We are grateful for the insightful feedback from the reviewers and have thoroughly revised our manuscript in line with your suggestions. Below, we detail how each concern and question has been addressed. We welcome any further feedback and kindly request your reconsideration of our revised submission, including a reassessment of its score if deemed appropriate.
>
> 1.  _"Further clarification is needed on how to interpret the findings from Figure 7 to better understand the implications of the results"_
>
> In our revised manuscript, we delve deeper into this aspect and provides novel insights in Figure 4 of Section 4.4. We specifically examined the accuracy disparities between the Vehicle and NUS-WIDE datasets, comparing these real-world discrepancies with the accuracy gaps observed in synthetic datasets split under identical $\alpha$ or $\beta$ parameters. This comparison reveals a strong positive correlation between the accuracy gaps in synthetic and real datasets. Specifically, synthetic datasets based on $\alpha$ exhibited a correlation coefficient of 0.83, while those based on $\beta$ demonstrated a coefficient of 0.39. This finding suggests that while not offering precise predictions, the performance on synthetic datasets with the same split parameters can provide a rough estimate of an algorithm's effectiveness on real VFL datasets.
>
> 2. _"Explore whether Spearman rank correlation can be replaced by the other mentioned correlations and, if possible, provide a comparison of their performance."_
>
> We have include the experiments of performance split by Pearson coefficient in Appendix G.10.
>
> > In comparing Spearman rank and Pearson correlation coefficients, we find Spearman rank effective for non-linear correlations, while Pearson is suitable for linear correlations. Figure 26 illustrates the performance differences on datasets split using these metrics.
> >
> >For linearly correlated datasets like gisette, Pearson's coefficient is preferable. It accurately captures the enhanced performance of C-VFL on highly correlated splits, a nuance Spearman rank-based splitting misses. Conversely, for non-linearly correlated datasets such as radar, Spearman rank-based correlation is more effective, highlighting variations in C-VFL performance that Pearson-based splits overlook. Therefore, the choice of correlation coefficient should align with the dataset's correlation characteristics.
>
> 3. _"Figure 5 indicates that three out of four VFL baselines are minimally impacted by alpha and beta"_
>
> To address concerns about the effect of $\alpha,\beta$, we introduced a new baseline, FedOnce, significantly influenced by split factors. For the detailed values of the performance shifts of other VFL algorithms, which may be overshadowed by C-VFL's variability, refer to Table 9. Here's a summary of how each algorithm is affected by $\alpha$ and $\beta$:
>
> - C-VFL: Performance varies up to 40% with $\alpha$ and up to 20% with $\beta$.
> - FedOnce: Shows a 10% variation with $\alpha$ on the letter dataset, and 5-7% on covtype, gisette, and epsilon; affected by 1-3% with $\beta$ on covtype and epsilon.
> - GAL: Impacted by 8% with $\alpha$ on letter and radar, and 2-5% on *all* other datasets; minimal impact from $\beta$.
> - SplitNN: Shows minimal variation due to robustness (though at a higher communication cost).
> - FedTree/SecureBoost: Not affected by feature split, theoretically matching centralized training performance.
>
> Notably, on the letter dataset, the performance ranking between FedOnce and GAL switches as $\alpha$ increases from 0.1 to 100. This suggests that GAL is better suited for imbalanced VFL scenarios, while FedOnce excels in balanced splits.
>
> This trend aligns with real VFL datasets: For the relatively imbalanced Vehicle dataset (small $\alpha$), GAL outperforms FedOnce by 2%. In contrast, on the NUS-WIDE dataset, both GAL and FedOnce show closely matched performance (within 0.3% difference), reinforcing that these synthetic dataset conclusions can effectively mirror real-world scenarios.
>
> 4. _"Communication efficiency can be moved to Appendix"_
>
> We have relocated the original Section 4.4 to the Appendix, noting that communication efficiency in VFL is largely unaffected by variations in feature split. Specifically, if each party possesses an equal number of features, the communication cost remains consistent. Moreover, even in scenarios where the number of features differs across parties, the impact on communication cost is negligible compared to the communication cost gap arising from distinct algorithmic designs.

---

> > ### Comment · Reviewer_oXEA · 2023-11-22
> >
> > Thanks for the response, I would like to raise my rating to 6.

---

### Official Review · Reviewer_MyAZ · 2023-11-01

**Soundness:** 2 fair
**Presentation:** 3 good
**Contribution:** 3 good
**Rating:** 6
**Confidence:** 3

**Summary:**

Because existing vertical federated learning methods either use randomly feature-partitioned dataset or a few public real-world datasets, this paper focuses on understanding the existing datasets’ characteristics (through dataset evaluation) and on enable wider feature distribution (through synthetic data construction). Specifically, it focuses on two factors — feature importance and feature correlation — and proposes evaluation metrics ($\alpha$ and $\beta$) and algorithms on how to split datasets according to these metrics. Given the proposed data construction methods, the authors compare existing VFL algorithms over the newly generated benchmarks and provide insights about their performance and communication efficiency.

**Strengths:**

- The paper focuses on an important yet often less-explored aspect of dataset evaluation and construction.
- The authors have provided algorithms to construct synthetic feature partitions covering a wider range of feature distribution than existing benchmarks.
- The insights obtained from VertiBench by comparing different split learning algorithms over different $\alpha$ and $\beta$ values are interesting.

**Weaknesses:**

- **The specific form of the correlation of two party correlation $\textrm{Pcor}$**. It’s not clear to me why the metric Pcor is defined as roughly speaking (upto an additional $\frac{1}{\sqrt{d}}$ constant) the standard deviation of the correlation matrix’s singular values. Consider a case where $m_i = m_j$ and the two parties have exactly the same set of features $X_i = X_j$ where the column features of $X_i$ are pairwise independent. In this case, the correlation matrix is the identity matrix $\textrm{cor}(X_i, X_j) = I_{m_i \times m_i}$ with singular values all being $1$. Then under the metric Pcor, these two parties' features would have 0 “correlation” because the singular values have zero variance while in reality there is indeed a somewhat strong notion of correlation (their features are identical). I would appreciate if the authors can provide an explanation of the reasoning in choosing this new metric and whether my suggested scenario would indeed be measured as having 0 “correlation” using pcor.

- **Exploring the performance variations of feature partitioning under the same $\alpha$ and $\beta$**. For party balance using $(\alpha_1, \ldots, \alpha_K)$, the feature partitioning procedure (algorithm 5) has internal randomness, thus making two runs of the same algorithm produce feature partitions that are likely different. In this case, I would expect to see some results on the performance distribution of the same split learning method over these different feature splits and understand how much variation there is. Similarly, I’m not sure if the argmin step (line 6) of Algorithm 1 could also yield multiple different solutions. If so, I think the authors should also explore the performances variation over these different partitions. Taking one step further, I wonder if there are algorithm ranking differences for the same $\alpha$ and $\beta$ but different feature partition realizations. If there are such ranking differences, then the authors should provide more detailed suggestions on how to use their benchmark (how many such splits should be used/averaged for such comparison). If there aren’t, I suggest the authors provide official splits after performing the sampling/optimization themselves so that future papers using these evaluation benchmarks would have a consistent comparable reporting.

- __Proposition 1__ can be trivially proved through taking logarithm of both sides of a telescoping product equality: $\frac{\mathcal{P}(y | X_K, \ldots, X_1)}{\mathcal{P}(y)} = \prod_{k=1}^K \frac{\mathcal{P}(y | X_k, \ldots, X_1)}{\mathcal{P}(y | X_{k-1}, \ldots, X_1)}$. However it’s not clear to me whether this likelihood ratio quantity really has deep connections to Shapley or the feature correlations, or is simply used as a heuristic quantity to motivate the story of feature balance and correlation.

- **Figure 2, 3, 4.** I appreciate the authors including Figure 2, 3, 4 in the paper. However, I believe they are basic sanity checks of the correctness of Theorem 1 and the correctness of the implementation of Algorithm 1, and do not provide additional new insights. I believe they should instead be put into the Appendix.

- __Communication efficiency__ Section 4.4 seems detached from the rest of the paper, as they do not explore in the axes of party balance ($\alpha$) and party correlation ($\beta$) explored in the rest of the paper.

- Minor typos: In Equation 1, the index should be $k$ instead of $i$. In Algorithm 5 (line 4), the $\cup$ should be $\cap$.

**Questions:**

- Can the authors provide further discussion of the scalability of their two partitioning methods? For example, how scalable is the computation of Shapley values used in the experiments (run-time as a function of the number of features)? How many partition dimensions can the optimization method BRKGA efficiently solve?
- Minor: Should the probablity $p_k$ (three lines above Theorem 1) be $r_k$?

---

> ### Author Response · Authors · 2023-11-21
> **Response to Reviewer MyAZ (Part 1)**
>
> We are grateful for the insightful feedback from the reviewers and have thoroughly revised our manuscript in line with your suggestions. Below, we detail how each concern and question has been addressed. We welcome any further feedback and kindly request your reconsideration of our revised submission, including a reassessment of its score if deemed appropriate.
>
> 1. _"The specific form of the correlation of two party correlation Pcor"_
>
> We have refined the definition of Icor in our revision, shifting from the original average of $\text{Pcor}(\mathbf{X}_i,\mathbf{X}_j)$ to averaging the relative difference $\text{Pcor}(\mathbf{X}_i,\mathbf{X}_j) - \text{Pcor}(\mathbf{X}_i,\mathbf{X}_i)$. This new approach leads to several key advancements:
>
> - Theoretical validation that **optimizing Icor results in ideal feature splits in optimal scenarios** (Theorem 1), consistent with our empirical findings on complex real datasets (see Figure 5 in Appendix D.2).
> - Provision of intuitive explanations for the design of Pcor metrics.
> - Visualization of the relationship between singular values, singular vectors, and inter-party correlation.
> - Theoretical demonstration that the range of Pcor is within $[0,1]$.
> - Theoretical proof of that Pcor and mcor (Taylor, 2020) are equivalent in measuring inner-party correlation.
> - Comprehensive re-evaluation of all experiments concerning correlation-based splits.
>
> Icor effectively measures the overall relative inter-party correlation within a global dataset. Previously, overlooking inner-party correlations of $\mathbf{X}_i$ and $\mathbf{X}_j$ led to discrepancies. High inner-party correlation typically results in increased Pcor values with other parties. The amendment to relative Pcor addresses this issue. For instance, consider two independent features $a_1, a_2$ forming two parties such that $\mathbf{X}_i=\mathbf{X}_j=[a_1,a_2]$. In this setup, $\text{Icor}(\mathbf{X}_i, \mathbf{X}_j)=0$, reflecting significant relative inter-party correlation (Icor is usually negative). Conversely, if we swap a feature to have $\mathbf{X}_i=[a_1,a_1], \mathbf{X}_j=[a_2,a_2]$, we get $\text{Pcor}(\mathbf{X}_i,\mathbf{X}_i)=1$, $\text{Pcor}(\mathbf{X}_j,\mathbf{X}_j)=0$, leading to $\text{Icor}(\mathbf{X}_i, \mathbf{X}_j)=-1$, indicative of a low inter-party correlation, aligning intuitively with expectations.
>
> 2. _"Exploring the performance variations of feature partitioning under the same  $\alpha$  and  $\beta$."_
>
> In our analysis presented in Figure 24 and 25, we investigated the convergence of both mean and variance in algorithm performance as the number of test repetitions increased. We observe both mean and variance converges after 5-10 runs. After the convergence, there are notable differences among the algorithms: C-VFL exhibited the largest variance, whereas SplitNN and FedTree maintained low variance levels. This study highlights the significant dependency of performance variance on the specific algorithm employed.
>
> It's important to note that variance in dataset splits is a characteristic to be leveraged, not a problem to be solved. Given that feature splits at similar importance or correlation levels are inherently non-unique, observing performance variances under identical split configurations can actually offer insights into the robustness of VFL algorithms. For instance, while SplitNN and FedTree demonstrate consistent robustness across various splits, C-VFL exhibits considerable variance in most scenarios.
>
> To facilitate a consistent comparison among algorithms, we recommend conducting at least 5-10 repeated experiments and reporting both the mean and variance of performances. Our observations indicate that five repetitions generally yield stable mean and variance for most evaluated methods. If an algorithm displays large variance after five runs, it probably suggests potential challenges in handling real-world feature splits with similar levels of importance or correlation.
>
> We also acknowledge and appreciate the suggestion for providing official splits to enable more consistent evaluations and rankings, recognizing its potential utility. We will provide them when releasing benchmark.

---

> ### Author Response · Authors · 2023-11-21
> **Response to Reviewer MyAZ (Part 2)**
>
> 3. _"Proposition 1: deep connections with Shapley value and feature correlation"_
>
> Proposition 1 intuitively links global data distribution with the importance and correlation among parties in machine learning models. The task of estimating this distribution, central to machine learning, poses significant challenges. Consequently, this complex relationship may not be directly translatable into mathematical terms like Shapley value or Pearson/Spearman rank correlation. Addressing the concern regarding Proposition 1's connection to Shapley values or feature correlations, it is essential to note that while the proposition is provable through mathematical manipulation, its role is more heuristic in nature. It serves as a guiding principle to understand the interplay between feature balance and correlation rather than indicating a deep, direct mathematical correlation with these concepts.
>
> 4. _"Figure 2, 3, 4 can be moved to Appendix"_
>
> We have moved Figure 2,3,4 and their related contents to Appendix D.2.
>
> 5.  _"Communication efficiency can be moved to Appendix"_
>
> We have relocated the original Section 4.4 to the Appendix, noting that communication efficiency in VFL is largely unaffected by variations in feature split. Specifically, if each party possesses an equal number of features, the communication cost remains consistent. Moreover, even in scenarios where the number of features differs across parties, the impact on communication cost is negligible compared to the communication cost gap arising from distinct algorithmic designs.
>
> 6. _"Can the authors provide further discussion of the scalability of their two partitioning methods?"_
>
>
> In Appendix D.3, we provide detailed timings for feature splits across datasets, encompassing a range of up to 0.58 million instances and 20,000 features. Additionally, we conducted scalability tests for the correlation-based split method on synthetic data of varying sizes.
>
> The importance-based method proves to be highly efficient, completing within 30 seconds across all datasets tested. This method, which randomly splits without necessitating the computation of Shapley values, still retains some Shapley-related properties, ensuring scalability for large datasets.
>
> Conversely, the correlation-based method, while not as rapid, is sufficiently efficient for practical applications. Our scalability tests demonstrate that this method can handle up to 10,000 features, with the time cost increasing approximately linearly with the number of features. This level of efficiency is adequate for most datasets typically used in VFL, such as realsim and CIFAR10.
>
> We emphasize that the permutation-based optimization in VertiBench is an orthogonal problem. Should more advanced optimization algorithms become available, they can be seamlessly integrated to enhance VertiBench's efficiency.

---

> > ### Comment · Reviewer_MyAZ · 2023-11-22
> > **Response to author rebuttal**
> >
> > I would like to thank the authors for their rebuttal and the changes they made to their paper. I’m generally satisfied with the response which have answered most of my questions. I have increased my score from 5 to 6.

---

### Author Response · Authors · 2023-11-22
**Summary of Reviewers' Concerns and Our Revised Contents**

We extend our deepest gratitude to the reviewers for their valuable advice. Below, we summarize the primary concerns raised and outline the revisions we've made in response:

1. **Insights on Correlation-Based Split** (Reviewer MyAZ):
   - Revision includes a redefined Icor metric, addressing the counterexample raised by Reviewer MyAZ.
   - Theoretical explanations, Theorem 1 and Propositions 2 and 3, align with empirical findings in Appendix D.2.
   - Intuitive explanation of the correlation variance provided, supported by visualizations in Figure 2.

2. **Feature Partitioning Variance** (Reviewer MyAZ):
   - Additional experiments in Section G.9 show convergence of both mean and variance in algorithm performance after 5-10 runs, illustrating that performance variance is an inherent property of the algorithm, reflective of its stability in real VFL scenarios.

3. **Scalability of Splitting Methods** (Reviewers MyAZ, 6Ygd):
   - Time costs detailed in Appendix D.3. Additional experiments demonstrate scalability in terms of dimensions and sample size.
   - Importance-based split method finishes within 30 seconds. Correlation-based split is efficient in handling large sample sizes (up to 10 million samples in under 4 minutes) and can accommodate up to 10,000 features in an hour.
   - Adding experiments to demonstrate that the number of parties hardly affect the efficiency under a fixed global dataset.

4. **Alignment of VertiBench to Real Scope** (Reviewer oXEA):
   - To enhance understanding, we have compiled comprehensive statistics from all datasets in Figure 4. Our analysis compares the relative performance on the Vehicle and NUS-WIDE datasets against synthetic datasets created to simulate these real-world splits using $\alpha$ or $\beta$. We observed **a positive correlation between the relative performance on synthetic datasets and real datasets for both $\alpha$ and $\beta$ parameters**. This finding indicates that VertiBench's synthetic datasets are effective in mirroring real-world data splits.

5. **Spearman Rank vs. Pearson Comparison** (Reviewer oXEA):
   - New experiments in Appendix G.10 comparing these metrics have been added.

6. **Impact of $\alpha$ and $\beta$ on Performance** (Reviewers oXEA, BWae):
   - Introduced a new baseline, FedOnce, showing significant influence by split factors.
   - Our analysis reveals that **three out of five** algorithms experience performance changes **up to 8% across multiple datasets** due to variations in these parameters. Notably, this variation **shifts the performance ranking** such as FedOnce and GAL. Conversely, SplitNN and FedTree demonstrate robustness to these split factors due to their inherent design.

7. **Independent Assumption Between Metrics and Choice Criteria** (Reviewers BWae, 6Ygd):
   - Guidance on metric selection provided in Section 2.1
   - Discussed the complexities of combining importance and correlation in feature split in Section 2.1 and further elaborated in Appendix H.1.

8. **Introduction of New Real Dataset** (Reviewer 6Ygd):
   - Developed "Satellite," a novel image-to-image VFL dataset, filling a gap in existing benchmarks. Details are in Appendix E, with evaluation in Appendix G.4.

We sincerely apologize for the last-minute submission, as extensive theoretical and empirical work was essential to adequately address the concerns. With the final 24 hours of rebuttal remaining, we humbly request the reviewers to consider our revised submission and notify us of any further concerns for a prompt response. Thank you for your time and consideration.

---

### Author Response · Authors · 2023-11-23
**Gratitude for Reviewer Feedback and Openness to Further Discussion**

We express our sincere thanks to Reviewer MyAZ and Reviewer oXEA for their prompt responses and thoughtful consideration of our revised manuscript. Your constructive feedback and the adjustment in ratings are immensely appreciated. With only **8 hours** remaining in the rebuttal period, we kindly encourage any other reviewers who may have additional feedback or concerns to please share them with us as soon as possible. Your insights are crucial for the further improvement of our work, and we are committed to addressing any additional points you may raise. Thank you once again for your valuable time and guidance.

---

### Meta-Review · Area_Chair_6B3h · 2023-12-10

**Metareview:**

This paper proposed method to generate synthetic data benchmarks for vertical federated learning. All reviewers agree this could be a useful contribution to the area of vertical federated learning.

Strength:
1. A remedy to address the lack of public datasets for vertical federated learning.
2. Show the synthetic data could be effective in mirroring real-world data splits.

Weakness:
1. While the synthetic data is useful, how it would help improve the development of VFL methods is still not clear.

**Justification For Why Not Higher Score:**

The approach appears to be a useful contribution for generating synthetic data, but its importance for the practical use is not particularly clear.

**Justification For Why Not Lower Score:**

All reviewers agreed that this paper is useful contribution.

---

### Decision · Program_Chairs · 2024-01-16

Accept (poster)